

**A Bayesian model to correct underestimated 3D wind speeds from sonic anemometers**
**increases turbulent components of the surface energy balance**
John M. Frank[1,2], William J. Massman[1], and Brent E. Ewers[2]
[1]U.S. Forest Service, Rocky Mountain Research Station, 240 W. Prospect Rd., Fort Collins, CO,

7  80526

[2]University of Wyoming, Department of Botany and Program in Ecology, 1000 E. University
Ave, Laramie, WY, 82071
*Corresponding author: telephone: +1 970 498 1319; email: jfrank@fs.fed.us
To be summited to Atmospheric Measurement Techniques



**Abstract**

19        Sonic anemometers are the principal instruments in micrometeorological studies of

turbulence and ecosystem fluxes. Recent studies have shown that common designs underestimate
vertical wind measurements because they lack a correction for transducer shadowing, with no
consensus on a suitable correction. We reanalyze a subset of data collected during field
experiments in 2011 and 2013 featuring two or four CSAT3 sonic anemometers. We introduce a
novel Bayesian analysis with the potential to resolve the three-dimensional correction by
optimizing differences between anemometers mounted both vertically and horizontally. A grid of
512 points (~ ±5° resolution in wind location) is defined on a sphere around the sonic
anemometer, from which the shadow correction for each transducer-pair is derived from a set of
138 unique state variables. Using the Markov chain Monte Carlo (MCMC) method, the Bayesian
model proposes new values for each state variable, recalculates the fast-response dataset,
summarizes the five-minute wind statistics, and accepts the proposed new values based on the
probability that they make measurements from vertical and horizontal anemometers more
equivalent. MCMC chains were constructed for three different prior distributions describing the
state variables: no shadow correction, the Kaimal correction for transducer shadowing, and
double the Kaimal correction, all initialized with 10% uncertainty. The final posterior correction
did not depend on the prior distribution and revealed both self- and cross-shadowing effects from
all transducers. After correction, the vertical wind velocity and sensible heat flux increased ~10%
with ~2% uncertainty, which was significantly higher than the Kaimal correction. We applied the
posterior correction to eddy covariance data from various sites across North America and found
that the turbulent components of the energy balance (sensible plus latent heat flux) increased on
average between 8-12%, with an average 95% credible interval between 6-14%. Considering this



is the most common sonic anemometer in the AmeriFlux network and is found widely within
FLUXNET, these results provide a mechanistic explanation for much of the energy imbalance at
these sites where all terrestrial/atmospheric fluxes of mass and energy are likely underestimated.





## 1. Introduction

The eddy-covariance technique has become the most commonly used method for

measuring the ecosystem exchange of mass and energy with the atmosphere. It is fundamental to
the global network of flux towers that are central to quantifying terrestrial carbon sinks and
sources (Baldocchi, 2003), to hydrological studies accounting for evapotranspiration and
sublimation (Biederman et al., 2014; Reba et al., 2012), and to the energy balance through the
turbulent fluxes of sensible and latent heat (Welch et al., 2015; Anderson and Wang, 2014).
There is a growing consensus within the flux community that many sonic anemometers, the core
instrument for all modern eddy-covariance systems, exhibit systematically biased underestimates
of the vertical wind component (Frank et al., 2016; Horst et al., 2015; Kochendorfer et al., 2012).
The ramifications for this are that all vertical fluxes (i.e., carbon dioxide, water vapor, latent
heat, sensible heat, momentum) are similarly underestimated for any ecosystem. This is roughly
consistent with the persistent energy balance closure problem across flux sites (Leuning et al.,
2012; Stoy et al., 2013; Wilson et al., 2002) where a vast majority are assumed to be systematic
biased towards low turbulent fluxes of sensible and latent heat.

Recent studies of Horst et al. (2015) and Frank et al. (2016) have shown that the error in

at least two non-orthogonal sonic anemometer designs can be traced to transducer shadowing
that remains uncorrected in the anemometer's firmware. In both studies, shadowing was
described a priori by theoretical formulations based on the wind-tunnel tests of Kaimal (1979),
yet there was no consensus on a correction. A shortcoming in the use of formulations derived for
single transducer-pairs in laminar flow to describe turbulent flow distortions around more
complex geometries (Fig. 1) is that shadowing between all transducers and structures cannot be
accurately represented or incorporated. A second problem is that in turbulent flow fields there



are few standards available to use as a calibration reference. Advancements in Bayesian
techniques (Gelman et al., 2004) have created the potential to resolve both of these issues by
incorporating prior knowledge of transducer flow distortions with a model that evaluates the
omnidirectionality of a sonic anemometer to produce a posterior 3D correction.
To quantify a 3D correction of the CSAT3 sonic anemometer, we reanalyze data from
field experiments conducted by Frank et al. (2013) and Frank et al. (2016) where wind
measurements from non-orthogonal anemometers mounted vertically and horizontally were
significantly different. We develop a Bayesian hierarchical model to evaluate three hypotheses:
(1) A 3D shadowing correction based solely on wind location can make a non-orthogonal
sonic anemometer omnidirectional.
(2) This correction increases vertical wind measurements more than expected from single
transducer shadowing because it accurately represents all shadowing between transducers.
(3) In ecosystems where these instruments are deployed, the application of this correction
will result in significantly higher turbulent components of the energy budget and improved
surface energy budget closure.

**2 Methods**
**2.1 Reanalysis of field experiments**
We reanalyze data from field campaigns conducted by Frank et al. (2013) and Frank et al.
(2016). To summarize them, experiments were conducted in 2011 and 2013 where multiple sonic
anemometers were deployed in a horizontal array at 24.5 m height on the Glacier Lakes
Ecosystem Experiments Site (GLEES) AmeriFlux scaffold above a subalpine forest in
southeastern Wyoming, USA (Frank et al., 2014). The anemometers were initially mounted





vertically, oriented west, arranged south to north, staggered up and down, and located 0.50 m
center-to-center from each other (Fig. 1). Periodically, some of the anemometers were rotated
90° around their *u*-axis and mounted horizontally. In this study we focus only on the CSAT3
sonic anemometer (Campbell Scientific, Inc., Logan, UT, USA) during times when both
vertically and horizontally mounted anemometers were present (Table 1). It is conventional to
describe the three dimensions of a sonic anemometer as the *u*, *v*, and *w*-axes. To reduce
confusion in describing horizontal anemometers, we refer to cardinal *u*, *v*, and *w* where the
measurements have been rotated to west-east (*u*), south-north (*v*), and down-up (*w*), which are
consistent with *u*, *v*, and *w* for vertically mounted anemometers. Finally, because our Bayesian
model is computationally intensive we reanalyze a subset of only 5% of the available data (see
section 2.3).
**2.2 The Bayesian model**

Bayesian statistics are based on Bayes theorem (Bayes and Price, 1763), which in modern

applications relates the posterior probability of a model parameter conditioned on data to the
product of the likelihood of the data and the prior probability of that parameter (Gelman et al.,
2004). In essence, the prior represents an initial educated guess or belief in the value of a model
parameter, the likelihood is the probability of observing the data if it was deterministically
generated from a model, and the posterior is an updated belief in the model parameter
considering each the prior, the model, and the data. Analytical evaluation of the posterior is
rarely possible, as is in our case, thus the posterior is commonly estimate through the Markov
chain Monte Carlo (MCMC) method, Gibbs sampling (Appendix A.1), and the Metropolis-
Hastings algorithm (Kruschke, 2010). The framework of our Bayesian model is to divide the
sphere around the sonic anemometer into approximately equal grid points and to define a prior



probability distribution of the 3D shadowing correction for each transducer pair at each location.
Then, the model proposes new corrections for each grid point, recalculates the fast-response
dataset, summarizes new five-minute wind statistics, determines the probability that the updated
measurements from vertical and horizontal anemometers are more equivalent using the proposed
correction versus the old one (i.e., the ratio of Eq. A13 evaluated for the proposed versus old
correction), and finally accepts/rejects the proposal probabilistically from this ratio to construct
the posterior correction. The model recursively adjusts the distribution that generates the
proposals to achieve between 25 and 50% acceptance rates. We define a grid of 512 points (~
±5° resolution of wind location) on a sphere around each of the three transducer pairs of the
sonic anemometer. Neglecting the upper and lower mounting arms that extend back into the
electronics housing and support block, the CSAT3 is symmetrical on either side of a transducer
pair, between the upper and lower hemispheres, and for each of the three transducer pairs. To
pool data and reduce computations, we make these assumptions of symmetry to describe all
1,536 points from a set of 138 unique state variables.

We test three prior corrections: no shadow correction, the Kaimal correction (Kaimal,

1979; Frank et al., 2016; Horst et al., 2015), and a doubling of the Kaimal correction (Frank et
al., 2016). The Kaimal correction is defined as $U_c = (1 - 0.16 + 0.16\theta/70)\acute{U}_C$ for $\theta \le 70°$ and
$U_c = \acute{U}_C$ for $\theta > 70°$, where $U_C$ and $\acute{U}_C$ are the measured and corrected wind velocities and $\theta$ is
the angle between the wind and the acoustic path.

The model predicts the standard deviation of the data, $\sigma_{f,i,c}$, during each five-minute

period, $f$, for each replicate sonic anemometer, $i$, in the three cardinal dimensions, $c$ (Fig. 1),
from a normal distribution with mean $\hat{\sigma}_{f,i,c}$ and standard deviation $\varepsilon$ (Eq. 1).
$$\sigma_{f,i,c} \sim N\big(\hat{\sigma}_{f,i,c}, \varepsilon^{-2}\big) \qquad\qquad (1)$$





The predicted mean is constructed in several steps. First, the state variable for the 3D correction,
$\vec{\alpha}_{T \times G}$, is a matrix representing each of the three transducer axes, $t$, for each grid point, $g$. Here it
does not matter if each grid point is independent or that they linked together through symmetry.
It is given a normal prior probability distribution with mean equal to the prior correction, $P_{t,g}$,
evaluated for each transducer-pair for wind blowing through the longitude, $\lambda$, and latitude, $\varphi$,
associated with each grid point with a predefined standard deviation equal to 0.1, or ±10%
uncertainty (Eq. 2).
$$\alpha_{T \times G_{t,g}} \sim \mathrm{N}\big(P_{t,g}, 0.1\big) \tag{2}$$

The 3D correction is applied to every 20-Hz sample, $j$, of the original measured wind velocity
data in transducer coordinates, $U_{T\,t,f,i,j}$. The multidimensional nominal predictor variable,
$\vec{x}_{G \times F \times I \times J_{g,f,i,j}}$, selects the corresponding grid point that occurs with every 20-Hz sample. The
corrected 20-Hz wind velocity in transducer coordinates is $\acute{U}_{T\,t,f,i,j}$ (Eq. 3).
$$\acute{U}_{T\,t,f,i,j} = U_{T\,t,f,i,j} \cdot \big(\vec{\alpha}_{T \times G} \vec{x}_{G \times F \times I \times J_{g,f,i,j}}\big) \tag{3}$$

The non-orthogonal data is transformed via matrix multiplication into orthogonal coordinates,
$\acute{U}_{S\,s,f,i,j}$, with the three sonic dimensions, $s$ (Eq. 4).
$$\acute{U}_{S\,s,f,i,j} = \vec{M}_{S \times T} \acute{U}_{T\,t,f,i,j} \tag{4}$$

The matrix, $\vec{M}_{S \times T}$, is specific to the CSAT3 geometry (Eq. 5).
$$\vec{M}_{S \times T} = \begin{bmatrix} -\frac{4}{3} & \frac{2}{3} & \frac{2}{3} \\ 0 & \frac{2}{\sqrt{3}} & -\frac{2}{\sqrt{3}} \\ \frac{2}{3\sqrt{3}} & \frac{2}{3\sqrt{3}} & \frac{2}{3\sqrt{3}} \end{bmatrix} = \begin{bmatrix} -1.333 & 0.667 & 0.667 \\ 0 & 1.155 & -1.155 \\ 0.385 & 0.385 & 0.385 \end{bmatrix} \tag{5}$$

In order for the model to predict data simultaneously from both vertical and horizontal
anemometers, a final corrected time series data set is produced in cardinal coordinates, $\acute{U}_{C_{f,i,c,j}}$.





$$\acute{U}_{C_{f,i,c,j}} = \left( \vec{x}_{F \times I \times J \times O_{f,i,j,o}} \vec{N}_{O \times C \times S} \right) \acute{U}_{S_{S,f,i,j}} \qquad (6)$$
The matrix $\vec{N}_{O \times C \times S}$ is straightforward (Eq. 7), and the multidimensional nominal predictor
variable, $\vec{x}_{F \times I \times J \times O_{f,i,j,o}}$, selects the orientation, $o$, of every 20-Hz sample.
$$\vec{N}_{O \times C \times S} = \begin{cases} \begin{bmatrix} 1 & 0 & 0 \\ 0 & 1 & 0 \\ 0 & 0 & 1 \end{bmatrix}, & o = 1 \ (i.e, vertical) \\ \begin{bmatrix} 1 & 0 & 0 \\ 0 & 0 & -1 \\ 0 & 1 & 0 \end{bmatrix}, & o = 2 \ (i.e., horizontal) \end{cases} \qquad (7)$$

161     Using the corrected time series data in cardinal coordinates, the model calculates the

average correction, $\beta_{F \times I \times C_{f,i,c}}$, for the five-minute standard deviation data for each anemometer
in each dimension (Eq. 8).
$$\beta_{F \times I \times C_{f,i,c}} = \frac{\sqrt{\frac{1}{J-1} \sum_{j=1}^{J} \left( \acute{U}_{C_{f,i,c,j}} - \frac{1}{J} \sum_{j=1}^{J} \acute{U}_{C_{f,i,c,j}} \right)^2}}{\sqrt{\frac{1}{J-1} \sum_{j=1}^{J} \left( U_{C_{f,i,c,j}} - \frac{1}{J} \sum_{j=1}^{J} U_{C_{f,i,c,j}} \right)^2}} \qquad (8)$$
Eq. 8 is equivalent to the ratio of the standard deviation of $\acute{U}_C$ divided by the standard deviation
of $U_C$ evaluated during each five-minute period for each sonic anemometer in each cardinal
dimension. The reference condition for every five-minute period, $\vec{\tilde{\sigma}}_{F \times C}$, is a state variable
representing the "true" standard deviation of wind velocity in each cardinal dimension. It is
assigned a uniform prior probability distribution that generously includes the "true" value by
allowing each $\tilde{\sigma}_{F \times C}$ to range from 0 to the maximum of all $U_C$ measurements (Eq. 9).
$$\tilde{\sigma}_{F \times C_{f,c}} \sim Unif\left(0, max(U_C)\right) \qquad (9)$$
Finally, the model predicts the mean for the standard deviation data as the reference divided by
the correction (Eq. 10).
$$\hat{\sigma}_{f,i,c} = \frac{\vec{\tilde{\sigma}}_{F \times C} \cdot \vec{x}_{F \times C_{f,c}}}{\vec{\beta}_{F \times I \times C} \cdot \vec{x}_{F \times I \times C_{f,i,c}}} \qquad (10)$$



The nominal predictor variable, $\vec{x}_{F \times C_{f,c}}$, selects the appropriate five-minute reference for each
cardinal dimension while the other nominal predictor variable, $\vec{x}_{F \times I \times C_{f,i,c}}$, selects the five-minute
correction for each sonic anemometer for that dimension.

To complete the Bayesian model definition, the model error is a state variable which is

assigned a prior probability distribution with a gamma distribution (Eq. 11).

$$\varepsilon \sim \mathrm{Gamma}(1, \acute{b}) \tag{11}$$

The variance of the gamma distribution, $\acute{b}$, is assigned the same variance as the prior distribution
for $\tilde{\sigma}_{F \times C}$ which is a uniform distribution (Eq. 12).

$$\acute{b} = \frac{\sqrt{12}}{max(U_C) - 0} \tag{12}$$

Distributions are defined where normal distributions are $\theta \sim N(a, b)$ with expected value $E(\theta) = a$
and variance $var(\theta) = 1/b^2$, gamma distributions are $\theta \sim Gamma(a, b)$ with $E(\theta) = a/b$ and $var(\theta)$
$= a/b^2$, and uniform distributions are $\theta \sim Unif(a, b)$ with $E(\theta) = (a+b)/2$ and $var(\theta) = (b-a)^2/12$.
**2.3 Analysis**

Our Bayesian analysis was conducted using R (version 3.2.2, (R Core Team, 2015))

within RStudio (version 0.99.486, (RStudio Team, 2015)). We constructed an MCMC chain of
10,000 steps for each of the three priors. Because the Bayesian model estimates are relative and
not an absolute correction (see discussion in section 4.1), we normalized each chain. This was
done in post-processing by dividing each update to the 138 unique corrections by the average
correction across all grid points. We inspected each chain and removed the first 500 steps for
burn-in and kept 1 out of every 140 steps to eliminate autocorrelation between steps for most
grid points (even after reducing to 138 state variables, a few of these were estimated from
relatively little data which unavoidably led to high autocorrelation between steps). This reduced
each MCMC chain to 68 steps, which we combined between the three priors to create a single





chain containing 204 independent samples of the posterior distribution of the normalized 3D
correction. To define an absolute correction such that equatorial measurements (i.e., $(u^2 + v^2)^{1/2}$)
are unchanged (see discussion in section 4.1), we applied the normalized correction to the time
series data of vertically mounted anemometers, calculated the corrected five-minute standard
deviations for equatorial winds, performed linear regression without an intercept (i.e., model the
average change in equatorial winds solely as a scaling factor) between these corrected and
uncorrected standard deviations for each of the 204 posterior samples, and determined the
scaling factor as the average of the 204 regression slopes. We divided all values in the
normalized 3D correction by this scaling factor to produce our final posterior correction.

Computation of the Bayesian model was extremely intensive: completion of the three

chains took upwards of two months of continuous computer processing (Windows 7, Intel®
Core™ i7-3630QM CPU @ 2.40 GHz processor, 1 TB solid state hard drive, 20 GB RAM).
During beta testing we attempted to estimate the 3D correction independently for all grid points
and all transducer pairs, with a single MCMC chain requiring a half-year to complete. Likewise,
we investigated increasing the number of grid points to obtain better resolution around the sphere
as well as increasing the amount of sonic anemometer data used from the Frank et al. (2013) and
Frank et al. (2016) datasets. In both cases we desired an order of magnitude better resolution or
more data, but the time required to complete a single MCMC chain quickly made these
improvements impractical. Instead, we determined that 512 grid points and 5% of the original
data was optimal considering these processing constraints.

There is a slight distinction to be made between the prior corrections which are defined as

a function, $\alpha(\lambda, \varphi)$, of the true longitude and latitude of the wind and the posterior correction
which is a function, $\alpha(\tilde{\lambda}, \tilde{\varphi})$, where ~ represents the uncorrected sonic anemometer measurement



of wind location. This means the posterior correction can be applied directly to the uncorrected
data whereas the prior should be applied recursively (i.e., determine the correction, update the
wind location, update the correction). To directly compare the prior and posterior corrections, we
also present our posterior correction with the wind locations recursively adjusted to approximate
the "true" longitude and latitude. For these analyses, we smoothed the posterior with a spherical
spline fit (Wahba, 1981) using R package mgcv (Wood, 2006).

We quantified the impact of shadowing on measurements of the standard deviations of

winds in the three dimensions and the sensible heat flux ($H$). This was done by applying the
posterior correction to the time series data of vertically mounted anemometers, calculating the
five-minute measurements, performing linear regression without an intercept between the
corrected and uncorrected measurements for each of the 204 posterior samples, and defining the
impact as a distribution composed of the 204 regression slopes. For $H$, the data was planar fit
rotated (Lee et al., 2004), time lag adjusted, and vapor flux corrected (Massman and Lee, 2002)
using ancillary data from the GLEES AmeriFlux site (Frank et al., 2014).

Finally, we quantified the impact of the 3D correction on the sum of the turbulent

components of the energy balance (i.e., sensible and latent ($LE$) heat flux) at various sites across
North America (Table 2). Each site featured a CSAT3, a fast-response hygrometer, and ancillary
meteorological data. Measurements of $LE$ were calculated similar to $H$ but including the Webb-
Pearman-Leuning correction (Webb et al., 1980). The impact of the 3D correction was quantified
as a distribution similar to above, except compiled from 30-minute time periods.

**3 Results**
**3.1 No correction**



244  Without any shadow correction applied, measurements between a vertically and a

245 horizontally mounted anemometer were different, which becomes clear when the variance

246 between two vertical anemometers is taken into account (Fig. 2b, d, f versus a, c, e). The root

247 mean square error (RMSE) in the 5-minute standard deviation of wind along all cardinal

248 dimensions ($u$, $v$, and $w$) combined was 9.4% between a vertical and a horizontal anemometer,

249 whereas the same metric between two vertical anemometers was 3.9%. The largest discrepancy

250 was along the cardinal $v$-axis, where the RMSE increased from 3.7% to 11.1% when comparing

251 vertical and horizontal anemometers (Fig. 2c versus d).

252 **3.2 The Kaimal prior correction**

253  The Kaimal correction is symmetrical with respect to each sonic transducer path (Fig. 3a,

254 c, e). Yet, the same correction when viewed in sonic coordinates reveals unique responses for $u$,

255 $v$, and $w$ (Fig. 3b, d, f). For small latitude winds, the corrections are small for $u$ and $v$

256 measurements, while those for $w$ are higher yet unstable around the equator (see discussion in

257 section 4.2). When the Kaimal correction was applied to the vertically mounted anemometers,

258 there were minor increases in the 5-minute standard deviations of $u$ and $v$ (0.8% and 2.9%) while

259 the increases for $w$ (5.6%) and $H$ (5.5%) were more substantial. This correction explained some

260 of the differences between vertically and horizontally mounted anemometers (Fig. 4) where the

261 RMSE for all cardinal dimensions combined was 6.2%, or 1.60 times greater than the same error

262 between two vertical anemometers. The discrepancy along the cardinal $v$-axis decreased to 6.6%,

263 or 1.86 times greater than the same error for two vertical anemometers, though some bias is still

264 apparent (Fig. 4c versus d). While the Kaimal correction is only one of three priors tested in our

265 Bayesian model, it is perhaps the most accepted algorithm currently available to correct

266 transducer shadowing in the CSAT3.



### 3.3 The Bayesian model

Figure 5 illustrates the approach of the Bayesian model. The model initializes the 512 grid points with a prior, in this case the Kaimal correction. No matter the transducer pair or vertical versus horizontal mounting, the 3D correction for all cases are identical but rotated versions of a common correction based on 138 unique state variables. For a single instantaneous wind, the simultaneous corrections for all six combinations of transducer pairs and mounting orientations will be different. As the MCMC chains progress, the Bayesian model will continuously adjust each of the 138 unique state variables so that measurements from the vertically and horizontally mounted anemometers are most similar based on the univariate conditional posterior probability distribution (Eq. A13). Much of the predictive power of the model comes from resolving the inconsistencies along the cardinal $v$-axis (Fig. 2d) where vertically and horizontally mounted anemometers are likely to be most dissimilar. Specifically, a vertically mounted CSAT3 should measure reasonably correct cross winds which must flow across the entire transducer and support structure of a horizontally mounted CSAT3.

Each MCMC chain was initialized with the mean of each prior, yet after convergence their posterior corrections were remarkably similar regardless of the choice of prior correction, with one peculiarity (Fig. 6). There was a clear linear relationship between the prior correction averaged across all 512 grid points (1.000 for no correction, 1.040 for the Kaimal correction, and 1.080 for the double-Kaimal correction) and the magnitude of the posterior correction (1.030, 1.064, and 1.098, respectively) that relates to the Bayesian model estimating a relative and not absolute correction (see discussion in section 4.1). The posterior correction is more than an estimate of the optimal solution, as it intrinsically accounts for the uncertainty of the correction at each of the 512 grid points (Fig. 7). Whereas each prior was defined with 10% uncertainty



(Eq. 2), much of the posterior correction has much lower standard deviations, especially around
the transducers where values were as low as 2.5% (Fig. 7a). These uncertainties can be expressed
in sonic coordinates for the $u$, $v$, and $w$ components, which in general show that the posterior
correction is most certain for winds along each of those axes, respectively (Fig. 7b-d), with the
uncertainty along the $w$ measurement ranging from 2.7-18.3%.

Figure 8 illustrates the completion of the Bayesian model where the same posterior

correction is applied to all transducer pairs and both mounting orientations. For every
instantaneous wind, application of these six different corrections ultimately results in the 5-
minute standard deviations of wind along the cardinal $u$, $v$, and $w$ axis being most similar
between the two mounting orientations.
**3.4 The posterior correction**

The posterior correction for each transducer pair is presented in Figure 9. These results

take into account the recursive adjustment to the wind locations and have been smoothed with a
spherical spline. Significantly more self-shadowing and cross-shadowing around the transducers
is visible than compared to the Kaimal prior (Fig. 9a, c, e versus Fig. 3a, c, e, in locations near all
transducers). These results are more certain (i.e., low standard deviations when compared to the
original 10% assigned to the prior) near the transducers, poorly constrained near the equator (Fig.
7a), and independent of the choice of prior correction (Fig. 6). Transforming the posterior
correction into sonic coordinates reveals that similar to the Kaimal prior, minimal $u$ and $v$
correction is required for small latitude winds (Fig. 9b, d versus 3b, d). But, the impact of the
additional transducer shadowing impacts $w$ measurements far more than was predicted (Fig. 9f
versus Fig. 3f) where the posterior was fairly certain for latitudes greater than $\pm13.5°$ (Fig. 7d);
the high uncertainty for near-equatorial wind is discussed in Sect. 4.2. The posterior corrected



CSAT3 was the most omnidirectional between vertically and horizontally mounted anemometers
(Fig. 10) where the RMSE for all cardinal dimensions combined was 5.3%, or 1.36 times greater
than the same error between two vertical anemometers. The discrepancy along the cardinal *v*-axis
was further reduced to 4.4%, which is only 1.20 times greater than the same error for two vertical
anemometers, and the bias has been removed (Fig. 10d versus 4d). When the posterior correction
was applied to the vertically mounted anemometers there were similar increases to the Kaimal
correction in the 5-minute standard deviations of *u* and *v* ($0.6 \pm 0.8$ [-1.0 2.2]%, $2.7 \pm 0.7$ [1.5
4.1]%, mean ± standard deviation [95% credible interval], Fig. 11a-b). But, compared to the
Kaimal correction, the increases in *w* ($10.6 \pm 1.7$ [7.6 13.9]%) and *H* ($9.9 \pm 1.6$ [7.2 12.6]%)
were substantial and significantly higher (Fig. 11c-d). We provide the MCMC chain for the final
posterior correction in the supplementary material as a tool for researchers to evaluate in other
sonic anemometer studies, to examine the uncertainty in ecosystem flux measurements, and to
investigate surface energy balance closure.
**3.5 Turbulent components of the ecosystem energy balance across a continent**
We applied the posterior correction to various sites across North America that deploy the
CSAT3 in their eddy-covariance instrumentation (Table 2). The estimated increase in *H* + *LE* at
these sites ranged from 8.1-11.6% with an average standard deviation and 95% credible interval
of ±1.9% and 6.1-13.8%. For all but one site, the increase in *H* + *LE* was significantly higher
than the increase due to the Kaimal correction. At the 2 m Yuma, AZ site, the lack of
significance is related to anomalously low instantaneous wind latitudes for which the *w*
correction is most uncertain (Fig. 7d).

**4 Discussion**





### 4.1 An omnidirectional standard

Perhaps the most important shortcoming in almost every sonic anemometer study is the

lack of a standard wind measurement to compare against. A fundamental problem is that the

principle of sonic measurements (Barrett and Suomi, 1949; Kaimal and Businger, 1963) involves

the observer effect, i.e. it is virtually impossible for sonic transducers to observe air parcels

without influencing them (Buks et al., 1998). Thus, any method that relies on a sonic

anemometer measurement as an absolute standard is flawed to an extent. And while we are

justified to believe that some sonic anemometer measurements are more accurate that others

(Frank et al., 2016) it is tenuous to choose any sonic anemometer measurement as a standard.

Then, what are the alternatives? Wind tunnels are extremely useful (Horst et al., 2015; van der

Molen et al., 2004) yet it is debatable that such laminar or quasi-laminar calibrations are

transferrable to turbulent field conditions (Hogstrom and Smedman, 2004). And, while other new

technologies such as Doppler Lidar exist (Sathe et al., 2011; Dellwik et al., 2015) their

application as a field reference standard has been limited.

What we address is the general problem of determining a calibration given an unknown

standard or nothing to compare against. Whether this problem exists in medicine (Lu et al.,

1997), acoustics (MacLean, 1940; Monnier et al., 2012), or micrometeorology with respect

calibrating sonic anemometry in turbulent flow fields, all approaches have a commonality of

testing the relative consistency of a response to unknown signals. In our situation, we hold the

3D sonic anemometer to an omnidirectional standard of relative consistency and contend that the

correction that best achieves this standard is statistically the most likely 3D calibration. A

CSAT3 without any 3D shadow correction is clearly not omnidirectional (Fig 2) as

measurements depend on the instrument's orientation. A CSAT3 with the Kaimal transducer





shadow correction is better at meeting this standard (Fig 4). However, the posterior 3D
correction is remarkably effective in making the CSAT3 omnidirectional (Fig. 10). Because the
posterior correction closely achieves the omnidirectional standard, we support our first
hypothesis and argue that it is the most accurate correction, in general, for the three dimensions
of the CSAT3. Whether or not the posterior correction reveals meaningful information regarding
vertical winds and turbulent fluxes is another matter discussed below.
A consequence of the omnidirectional standard is that implicitly this produces only
relative results. Indeed, our Bayesian posterior has no meaning in an absolute sense without the
additional constraint that equatorial winds should be unchanged by the correction. We do not
specify the 3D correction at any of the grid points nor we do we specify a reference or "true"
condition for the standard deviation of wind during any five minute period. Because of this, the
parameter estimates for $\vec{\sigma}_{F \times C}$ and $\vec{\alpha}_{T \times G}$ only have meaning relative to each other. This issue is
confounded by the choice of prior distributions which vary dramatically in shape, but produce
similar posteriors except for differences in their absolute magnitudes (Fig. 6), i.e., higher
magnitude priors produce higher magnitude posteriors. Which absolute magnitude is correct?
Without specifying an absolute standard, the answer is none of them. To facilitate comparison
and combination of the posteriors we normalized the three MCMC chains.
There is a clear need to specify an absolute standard to reference our results. Without
one, our normalized posterior correction reduced the 5-minute standard deviations for equatorial
winds (i.e., the *u-v* plane) by 7%. Does this make physical sense? No. The idea that equatorial
winds should not be changed is consistent with the expectation that the CSAT3 measures
accurate equatorial winds, something that has been demonstrated in both wind tunnels and field
campaigns (Yahaya and Frangi, 2004; Friebel et al., 2009). Even the Kaimal correction, which is
an absolute correction, predicts <0.1% error in our measurements of equatorial winds. Because
the omnidirectional standard is only relative, we impose an additional absolute standard by
defining the average correction for equatorial winds to be zero, which is simply achieved by
scaling the normalized posterior correction by 7%. While there certainly is some leeway in this
constraint, if the normalized posterior correction were scaled by anything other than $7 \pm 1.4\%$
then the correction to horizontal winds would be significantly different (95% credible interval)
than both zero and the Kaimal correction (Fig. 11a-b) and would run counter to our belief that
the CSAT3 measures reasonable accurate horizontal winds.
**4.2 Impact on vertical wind measurements and sensible heat flux**

Recent studies have questioned the accuracy of CSAT3 vertical wind velocity

measurements (Frank et al., 2013; Kochendorfer et al., 2012) culminating with Horst et al.
(2015) and Frank et al. (2016) who identified the anemometer's lack of transducer shadowing
correction as the root cause. Quantifying the inaccuracy and determining how to fix this problem
has been a challenge. While each of these studies estimated different errors in $w$ at their field
sites (3.5% (Horst et al., 2015), 6-10% (Frank et al., 2013), 5.5-12.5% (Frank et al., 2016), and
14% (Kochendorfer et al., 2012)), it wasn't until Horst et al. (2015) proposed the application of
the Kaimal correction (Kaimal, 1979) that a mechanistic explanation was used to quantify the
underestimate. Whether or not the Kaimal correction is sufficient is a matter of debate, but it
currently represents the best prior knowledge to explain the CSAT3's shortcomings.

Solely because the posterior correction makes the CSAT3 more omnidirectional does not

imply that field measurements of vertical wind and turbulent fluxes are impacted, nor does this
assure that these impacts would be due to anything more than chance. Even with the uncertainty
in the posterior $w$ correction explicitly quantified (Fig. 7d) it is difficult to foresee if $w$ is





significantly impacted without applying the posterior correction to actual data. A powerful
attribute of the Bayesian analysis is that the posterior correction can be applied to raw data to
produce probability distribution estimates for $w$ and $H$ from which statistical inferences can be
made. Using GLEES data, Fig. 11c-d confirms that to achieve an omnidirectional sensor (Fig.
10) with minimal change to horizontal winds (Fig. 11a-b) the required correction will increase
both $w$ and $H$ by an average of 10.6% and 9.9%, which is significantly more (>95% credible
interval) than predicted by the Kaimal prior. We argue that this significant increase in the vertical
wind occurs because the posterior correction more accurately accounts for all shadowing
between transducers (Fig. 9 versus Fig. 3), therefore we support our second hypothesis.
Also of note, there are instabilities in the prior and posterior $w$ corrections for near-
equatorial winds that occur at latitudes less than ±4° (6 inflection points around the equator, Fig.
3f and 9f). The mathematical cause for these instabilities and the locations of the inflection
points are derived in Appendix A.2, and unless the corrections for the three transducers are
exactly equal everywhere around the equator these instabilities will exist. The existence of these
instabilities should cause concern for eddy-covariance measurements. The ultimate impact of this
phenomena is difficult to know, because on one hand, $w$ for latitudes less than ±4° are by
definition very small, but on the other, these eddies constitute a large proportion of winds that
exist under field conditions and their correction is currently unpredictable. For example, at
GLEES 30% of winds occur at latitudes within ±4° (unpublished analysis of Figure 4 from Frank
et al. (2016)). It is unknown how aggressively the correction for these winds approaches ±∞ or if
more inflection points actually occur. For all non-orthogonal geometries, not just the CSAT3, if
any transducer shadowing occurs at the equator, there will be instabilities in the $w$ correction.
**4.4 Impact across global flux networks**





Energy balance is a fundamental ecosystem concept where the flow of available energy
into an ecosystem influences the microclimate, drives photosynthesis, and establishes trophic
levels among the biota (Odum, 1957; Fisher and Likens, 1973; Teal, 1962). Yet, eddy covariance
studies of ecosystem fluxes seldom delve into details of energy flow beyond the generation of
sensible and latent heat. It is often stated that most eddy covariance sites underestimate these
turbulent components of the energy balance by 10-20% when compared to the available energy
(Wilson et al., 2002; Foken, 2008; Stoy et al., 2013; Leuning et al., 2012; Franssen et al., 2010).
Even when sites thoroughly account for lesser components such as energy stored in the biomass
or canopy air, the turbulent energy can still be 1-14% underestimated (Heilman et al., 2009;
Oliphant et al., 2004; Barr et al., 2006; Wang et al., 2012). It is common for sites to deal with
this problem by forcing energy balance closure by increasing $H$ and/or $LE$ (Heilman et al., 2009;
Oliphant et al., 2004; Twine et al., 2000; Scott et al., 2004) or even carbon fluxes (Barr et al.,
2006) by the percent of the energy imbalance. Is there a mechanistic reason why so many sites
believe their turbulent fluxes are underestimated? While it is difficult to generalize for every site,
one similarity among these studies (Heilman et al., 2009; Oliphant et al., 2004; Barr et al., 2006;
Wang et al., 2012; Twine et al., 2000; Scott et al., 2004) is they all feature a CSAT3, as do ~60%
of all sites in the AmeriFlux network (unpublished summary of 150 the 228 sites where
anemometer information was available, list accessed at http://ameriflux.lbl.gov/ in November
2015) and numerous sites distributed across the world within FLUXNET
(http://fluxnet.fluxdata.org/).
After applying the posterior correction to the CSAT3 at our site, measurements of one of
the energy balance components, $H$, increased $9.9 \pm 1.6\%$, which is about twice the 5.5% increase
predicted the Kaimal correction (Fig. 11) (note, the field experiments were conducted without a



co-located fast-response hygrometer, hence we do not estimate the impact on *LE* at our site).
However, we must consider that our field site in Wyoming is unusual, with extreme wind and
turbulence, and where summer friction velocity ($u_*$) averages 0.6 m s$^{-1}$ (Frank et al., 2016).
While this made GLEES a good location to conduct the turbulent field experiments that led to
the development of the posterior correction, do our results lead to similar impacts on ecosystem
fluxes elsewhere? To answer this we applied the posterior correction to eddy covariance
measurements at various sites across North America that employ the CSAT3 (Table 2). We
found that the sum of the turbulent components of the energy balance (sensible plus latent heat
flux) increased on average between 8-12% with the average 95% credible interval being 6-14%.
At most sites this was significantly higher than applying the Kaimal correction. Thus, it is highly
probable that at flux sites that employ the CSAT3 sonic anemometer the posterior correction will
significantly increase the turbulent components of the energy budget and explain much of the
ubiquitous energy imbalance problem; therefore we support our third hypothesis.
Are the results from this study applicable to the non-orthogonal sonic anemometers
produced by other manufacturers? Possibly. Frank et al. (2016) showed that the Applied
Technologies, Inc. A-probe shares a similar transducer geometry, a lack of a shadow correction
algorithm, and similar differences between vertically and horizontally mounted anemometers, so
it would be reasonable to expect a similar 3D correction for that instrument. But other
manufacturers do apply wake corrections in their firmware that are traceable to wind tunnel
calibrations. Are these adequate? Maybe not, as non-orthogonal anemometers from other
manufacturers have been implicated to erroneously measure the vertical wind (Kochendorfer et
al., 2012; Nakai et al., 2014; Nakai and Shimoyama, 2012). Without details of the calibrations or
the wake corrections it is difficult to know. Regardless, for any non-orthogonal sonic





anemometer with vertically oriented transducers, equatorial instabilities are likely to exist
(Appendix A.2) that would be extremely difficult to characterize with only a series of wind
tunnel calibrations. One benefit of our methodology is that it allows an independent check on the
sufficiency of these wake corrections. If such an instrument fails to consistently measure 3-
dimensional winds (i.e., it responds like Fig. 2), then our methodology would estimate a
posterior correction that could correct a wake-corrected anemometer. Because ~90% of all
AmeriFlux sites use non-orthogonal sonic anemometers (Frank et al., 2013; Nakai et al., 2014), it
would be appropriate to investigate this issue for all non-orthogonal sonic anemometer designs.
**4.5 The next step**

While these results reveal much about the nature of shadowing in a non-orthogonal sonic

anemometer, there is much more to be done. First, due to the intense computational burden of
this analysis we never fully utilized our data. While we only analyzed 5% of the available data,
limited the 3D correction to approximately $\pm 5°$ resolution and only 138 unique corrections, and
terminated the Bayesian MCMC chains after only 10,000 steps, it still took months of continuous
processing with extensive memory usage to produce these results. Obviously there is an
opportunity to adapt this analysis to run on multiple cores or a supercomputer. As we developed
our analysis it became apparent that with more data the standard deviations of the posterior
distribution improved; we foresee that with 20 times more data the uncertainty in the posterior
correction would be further reduced. Adaptation to a high-performance computer will allow for a
more precise grid, longer MCMC chains, and a lower standard deviation of the posterior
distribution.

Our results draw extensively on the symmetry of the CSAT3, which fails to account for

the upper and lower mounting arms that extend back into the electronics housing and support




block. We beta tested our model to solve for the 3D correction independently for each transducer
and for all grid points around the sphere. We abandoned this because winds at GLEES are fairly
unidirectional causing many of the grid points to be poorly characterized. Plus with an order of
magnitude more unique grid points to solve, the computation took over 5 months to complete
just one MCMC chain! There is a middle ground between assuming symmetry and pooling data,
i.e., the correction for the A transducer pair could be considered symmetrical along the $u$-$w$ plane
and the corrections for transducer pairs B and C are mirror images of each other. In addition to
solving the problem with less assumptions of symmetry, more experimental manipulations
should be tested. We only tested a 90° rotation along the $u$-axis, but there are limitless other
manipulations that would help characterize the shadowing around the entire 3D space
surrounding an anemometer. Our model could easily be adapted to handle different
manipulations using Eq. (7). This equation can be expanded to account for a limitless number of
manipulations within the same analysis.
Our results using the posterior correction (Fig. 10) show that there is still unexplained
residual error, though we expect some of this to be reduced with our suggestions above. While
Horst et al. (2015) showed that to a first order that transducer shadowing is a function of the
longitude and latitude of the instantaneous wind, the impact of other covariates such as wind
velocity and turbulence may need to be considered. An advantage of performing our analysis in a
Bayesian framework is that the model can be expanded to incorporate the effects of these
covariates.
And finally, our posterior correction and methodology should be compared to other
independent analysis of sonic anemometer shadowing such as wind tunnel data (Horst et al.,
2015) or an independent Doppler Lidar system (Sathe et al., 2011). Care should be taken when



incorporating these results, as anemometers could respond differently under laminar flow in a
wind tunnel versus under turbulent field conditions. Regardless, a key to resolving this problem
will be to embrace new technologies, new experimental designs, and new analyses.

**5 Conclusion**

The non-orthogonal CSAT3 sonic anemometer produces different results (Fig. 2) when it

is mounted horizontally instead of vertically (Fig. 1). Assuming that the primary source of this
error is shadowing across the various transducers, a Bayesian model can estimate a posterior
correction (Fig. 8) that ultimately makes measurements from vertically and horizontally mounted
anemometers most similar (Fig. 10). Even when taking into account the uncertainty of the
posterior correction (Fig. 7) the increases in vertical wind velocity and sensible heat flux
measurements are significantly larger and are approximately twice the magnitude of the Kaimal
correction (Fig. 11). When this posterior correction is applied to various eddy covariance sites
across North America, the turbulent components of the ecosystem energy balance (sensible plus
latent heat flux) increased between 8.1-11.6%, with an average 95% confidence that this increase
was between 6.1-13.8% (Table 2). Considering this is the most common sonic anemometer in the
AmeriFlux network and is found in all the regional networks that comprise FLUXNET, these
results have major implications for countless studies that use the eddy-covariance technique to
measure terrestrial/atmospheric exchange of mass and energy.

**Acknowledgments**

We thank Jorge Ramirez, Susan Howe, Mario Bretfeld, Kelly Elder, Banning Starr, Bill

Kustas, and Joe Alfieri for providing data from their unique field sites. We especially thank Ben



Bird for his countless hours of statistical advice in developing the Bayesian model. This study
was funded by the U.S. Forest Service, the Wyoming Water Development Commission, the
USGS, the NSF (awards EPS-1208909 and EAR-0444053), and the DOD Army Research Office
(W911NF-05-1-0558 and W911NF-05-1-0126).

**Appendix**
**A.1 Univariate conditional posterior distribution functions for Gibbs sampling**
For the univariate conditional posterior distribution functions there is a distinction
between independent grid points versus those linked together through symmetry. In the case of
the former, these functions can be evaluated for each unique grid point, $g$, for each transducer
pair, $t$. In the case of the latter, $g$ and $t$ refer to the sets of all grid points and transducers that
share the same unique state variable for their shadow correction, and these functions can be
applied to each of these unique sets.
First, using Bayes Theorem, the joint posterior distribution of the model parameters can
be expressed as being proportional to the product of the likelihood of the data and the joint prior
distribution of the model parameters (Eq. A1).
$$p\left(\tilde{\sigma}_{F\times C_{f,c}}, \alpha_{T\times G_{t,g}}, \varepsilon \middle| \sigma_{f,i,c}\right) \propto p\left(\sigma_{f,i,c} \middle| \tilde{\sigma}_{F\times C_{f,c}}, \alpha_{T\times G_{t,g}}, \varepsilon\right) p\left(\tilde{\sigma}_{F\times C_{f,c}}, \alpha_{T\times G_{t,g}}, \varepsilon\right) \text{(A1)}$$
Because the prior distributions for three model parameters are independent, the joint prior
distribution can be written as the product of the individual probabilities (Eq. A2).
$$p\left(\tilde{\sigma}_{F\times C_{f,c}}, \alpha_{T\times G_{t,g}}, \varepsilon \middle| \sigma_{f,i,c}\right) \propto p\left(\sigma_{f,i,c} \middle| \tilde{\sigma}_{F\times C_{f,c}}, \alpha_{T\times G_{t,g}}, \varepsilon\right) p\left(\tilde{\sigma}_{F\times C_{f,c}}\right) p\left(\alpha_{T\times G_{t,g}}\right) p(\varepsilon)$$
$$\text{(A2)}$$
The likelihood of the data is normally distributed (Eq. A3).
$$p\left(\sigma_{f,i,c} \middle| \tilde{\sigma}_{F\times C_{f,c}}, \alpha_{T\times G_{t,g}}, \varepsilon\right) = \frac{1}{\sqrt{2\pi}\varepsilon} e^{\left(-\frac{1}{2\varepsilon^2}\left(\sigma_{f,i,c}-\hat{\sigma}_{f,i,c}\right)^2\right)} \qquad \text{(A3)}$$





Because $\hat{\sigma}_{f,i,c}$ is both a function of $\tilde{\sigma}_{F\times C_{f,c}}$ and $\alpha_{T\times G_{t,g}}$ the likelihood is indeed a function of all
three model parameters. The individual prior distributions for $\tilde{\sigma}_{F\times C_{f,c}}$, $\alpha_{T\times G_{t,g}}$, and $\varepsilon$ are
uniformly (Eq. A4), normally (Eq. A5), and gamma (Eq. A6) distributed, respectively.
$$p\left(\tilde{\sigma}_{F\times C_{f,c}}\right) = \begin{cases} \frac{1}{max(U_C)}, & 0 \le \tilde{\sigma}_{F\times C_{f,c}} \le max(U_C) \\ 0, & otherwise \end{cases} \tag{A4}$$
$$p\left(\alpha_{T\times G_{t,g}}\right) = \frac{1}{\sqrt{2\pi}(0.1)} e^{\left(-\frac{1}{2(0.1)^2}\left(\alpha_{T\times G_{t,g}} - P_{t,g}\right)^2\right)} \tag{A5}$$
$$p(\varepsilon) = \acute{b}e^{-b} \tag{A6}$$
Gibbs sampling for each model parameter is based on the univariate conditional posterior
distribution which assumes that all other model parameters plus the data are given (in the case of
sampling within a multidimensional array, all other parameters within that array are given except
the one at the index being evaluated). For $\tilde{\sigma}_{F\times C_{f,c}}$ the univariate conditional posterior distribution
can be expressed as a form of Bayes Theorem (Eq. A7).
$$p\left(\tilde{\sigma}_{F\times C_{f,c}}\Big|\underline{\tilde{\sigma}_{F\times C}}_{-f,c}, \underline{\alpha_{T\times G}}, \varepsilon, \underline{\sigma}\right) = \frac{p\left(\underline{\tilde{\sigma}_{F\times C}}, \underline{\alpha_{T\times G}}, \varepsilon\big|\underline{\sigma}\right) p(\underline{\sigma})}{p\left(\underline{\tilde{\sigma}_{F\times C}}_{-f,c}, \underline{\alpha_{T\times G}}, \varepsilon, \underline{\sigma}\right)} \tag{A7}$$
The under-bar denotes all elements within a multidimensional array, while the notation $\underline{\tilde{\sigma}_{F\times C}}_{-f,c}$
means all elements of $\vec{\tilde{\sigma}}_{F\times C}$ except for $\tilde{\sigma}_{F\times C_{f,c}}$. On right side of Eq. A7, both the second term in
the numerator and the denominator are assumed given and can be omitted if the equal sign is
changed to a proportional sign. The first term in the numerator, $p\left(\underline{\tilde{\sigma}_{F\times C}}, \underline{\alpha_{T\times G}}, \varepsilon\big|\underline{\sigma}\right)$, is the joint
posterior distribution summed across all parameters (Eq. A8).
$$p\left(\underline{\tilde{\sigma}_{F\times C}}, \underline{\alpha_{T\times G}}, \varepsilon\big|\underline{\sigma}\right) \propto$$
$$\prod_{f=1}^{F}\prod_{c=1}^{3}\left\{\left[\prod_{i=1}^{I} p\left(\sigma_{f,i,c}\Big|\tilde{\sigma}_{F\times C_{f,c}}, \alpha_{T\times G_{t,g}}, \varepsilon\right)\right] p\left(\tilde{\sigma}_{F\times C_{f,c}}\right)\right\} \prod_{t=1}^{3}\prod_{g=1}^{G} p\left(\alpha_{T\times G_{t,g}}\right) p(\varepsilon) \tag{A8}$$



Assuming that all but $\tilde{\sigma}_{F \times C_{f,c}}$ is given plus requiring that the proposed value for $\tilde{\sigma}_{F \times C_{f,c}}$ is
within the valid range (i.e., $p\left(\tilde{\sigma}_{F \times C_{f,c}}\right)$ is constant and can be omitted) Eq. A7 simplifies to Eq.
A9.

$$p\left(\tilde{\sigma}_{F \times C_{f,c}} \Big| \underline{\tilde{\sigma}_{F \times C}}_{-f,c}, \underline{\alpha_{T \times G}}, \varepsilon, \underline{\sigma}\right) \propto \prod_{i=1}^{I} p\left(\sigma_{f,i,c} \Big| \tilde{\sigma}_{F \times C_{f,c}}, \alpha_{T \times G_{t,g}}, \varepsilon\right) \quad \text{(A9)}$$

Substituting in the likelihood from Eq. A3 and simplifying gives the univariate conditional
posterior distribution for $\tilde{\sigma}_{F \times C_{f,c}}$ (Eq. A10).

$$p\left(\tilde{\sigma}_{F \times C_{f,c}} \Big| \underline{\tilde{\sigma}_{F \times C}}_{-f,c}, \underline{\alpha_{T \times G}}, \varepsilon, \underline{\sigma}\right) \propto e^{\left(-\frac{1}{2\varepsilon^2} \sum_{i=1}^{I} \left(\sigma_{f,i,c} - \hat{\sigma}_{f,i,c}\right)^2\right)} \quad \text{(A10)}$$

The univariate conditional posterior distribution for $\alpha_{T \times G_{t,g}}$ can be expressed as Bayes Theorem
(Eq. A11).

$$p\left(\alpha_{T \times G_{t,g}} \Big| \underline{\tilde{\sigma}_{F \times C}}, \underline{\alpha_{T \times G}}_{-t,g}, \varepsilon, \underline{\sigma}\right) = \frac{p\left(\underline{\tilde{\sigma}_{F \times C}}, \underline{\alpha_{T \times G}}, \varepsilon \Big| \underline{\sigma}\right) p(\underline{\sigma})}{p\left(\underline{\tilde{\sigma}_{F \times C}}, \underline{\alpha_{T \times G}}_{-t,g}, \varepsilon, \underline{\sigma}\right)} \quad \text{(A11)}$$

Again, only the first term in the numerator must be evaluated while assuming that all but $\alpha_{T \times G_{t,g}}$
are given (Eq. A12).

$$p\left(\alpha_{T \times G_{t,g}} \Big| \underline{\tilde{\sigma}_{F \times C}}, \underline{\alpha_{T \times G}}_{-t,g}, \varepsilon, \underline{\sigma}\right) \propto \prod_{f=1}^{F} \prod_{i=1}^{I} \prod_{c=1}^{3} p\left(\sigma_{f,i,c} \Big| \tilde{\sigma}_{F \times C_{f,c}}, \alpha_{T \times G_{t,g}}, \varepsilon\right) p\left(\alpha_{T \times G_{t,g}}\right)$$

(A12)

Substituting in both the likelihood of the data (Eq. A3) and the prior distribution for $\alpha_{T \times G_{t,g}}$ (Eq.
A5) and simplifying yields the univariate conditional posterior distribution for $\alpha_{T \times G_{t,g}}$ (Eq. A13).

$$p\left(\alpha_{T \times G_{t,g}} \Big| \underline{\tilde{\sigma}_{F \times C}}, \underline{\alpha_{T \times G}}_{-t,g}, \varepsilon, \underline{\sigma}\right) \propto e^{\left(-\frac{1}{2\varepsilon^2} \sum_{f=1}^{F} \sum_{i=1}^{I} \sum_{c=1}^{3} \left(\sigma_{f,i,c} - \hat{\sigma}_{f,i,c}\right)^2 - \frac{1}{2(0.1)^2}\left(\alpha_{T \times G_{t,g}} - P_{t,g}\right)^2\right)}$$

(A13)



An important issue is that $\hat{\sigma}_{f,i,c}$ is a function of $\alpha_{T \times G_{t,g}}$ and must be evaluated for every proposed
change to the 3D correction. This is computationally intensive and causes a bottleneck in the
analysis. Finally, the univariate conditional posterior distribution for $\varepsilon$ can be expressed as Bayes
Theorem (Eq. A14).
$$p\left(\varepsilon \middle| \underline{\tilde{\sigma}_{F \times C}}, \underline{\alpha_{T \times G}}, \underline{\sigma}\right) = \frac{p\left(\underline{\tilde{\sigma}_{F \times C}}, \underline{\alpha_{T \times G}}, \varepsilon \middle| \underline{\sigma}\right) p(\underline{\sigma})}{p\left(\underline{\tilde{\sigma}_{F \times C}}, \underline{\alpha_{T \times G}}, \underline{\sigma}\right)} \tag{A14}$$

Only the first term in the numerator must be evaluated while assuming that all but $\varepsilon$ are given
(Eq. A15).
$$p\left(\varepsilon \middle| \underline{\tilde{\sigma}_{F \times C}}, \underline{\alpha_{T \times G}}, \underline{\sigma}\right) \propto \prod_{f=1}^{F} \prod_{i=1}^{I} \prod_{c=1}^{3} p\left(\sigma_{f,i,c} \middle| \tilde{\sigma}_{F \times C_{f,c}}, \alpha_{T \times G_{t,g}}, \varepsilon\right) \tag{A15}$$

Substituting in the likelihood from Eq. A3 and simplifying yields the univariate conditional
posterior distribution for $\varepsilon$ (Eq. A16)
$$p\left(\varepsilon \middle| \underline{\tilde{\sigma}_{F \times C}}, \underline{\alpha_{T \times G}}, \underline{\sigma}\right) \propto \varepsilon^{-3FI} e^{\left(-\frac{1}{2\varepsilon^2} \sum_{f=1}^{F} \sum_{c=1}^{3} \sum_{i=1}^{I} (\sigma_{f,i,c} - \hat{\sigma}_{f,i,c})^2\right)} \tag{A16}$$

**A.2 Instability in the *w* correction for near equatorial winds**

For a CSAT3, the amount of correction applied to the vertical wind velocity, expressed as

the individual corrections $\alpha_A(\lambda, \varphi)$, $\alpha_B(\lambda, \varphi)$, and $\alpha_C(\lambda, \varphi)$ for the three transducer pairs $A$, $B$,
and $C$ as functions of longitude, $\lambda$, and latitude, $\varphi$, is:
$\frac{w_{corrected}}{w_{uncorrected}} = \frac{2}{3\sqrt{3}} \left[ \left(-\frac{\cos\lambda}{2\tan\varphi} + \frac{\sqrt{3}}{2}\right) \alpha_A(\lambda, \varphi) + \left(\frac{\cos\lambda + \sqrt{3}\sin\lambda}{4\tan\varphi} + \frac{\sqrt{3}}{2}\right) \alpha_B(\lambda, \varphi) + \left(\frac{\cos\lambda - \sqrt{3}\sin\lambda}{4\tan\varphi} + \right.$
$\left. \frac{\sqrt{3}}{2}\right) \alpha_C(\lambda, \varphi) \right]$                                   (A17)
If the individual corrections for the three transducer pairs never approach 0 or $\pm\infty$, which is a
safe assumption considering they are always around 1 (Figs. 3a, c, e and 9a, c, e), the limit of this
as the latitude approaches the equator is:





$$\lim_{\varphi \to 0} \frac{w_{corrected}}{w_{uncorrected}} = \frac{1}{3}\big(\alpha_A(\lambda,\varphi) + \alpha_B(\lambda,\varphi) + \alpha_C(\lambda,\varphi)\big) +$$
$$\frac{2}{3\sqrt{3}}\left[\left(-\frac{\cos\lambda}{2}\right)\alpha_A(\lambda,\varphi) + \left(\frac{\cos\lambda+\sqrt{3}\sin\lambda}{4}\right)\alpha_B(\lambda,\varphi) + \left(\frac{\cos\lambda-\sqrt{3}\sin\lambda}{4}\right)\alpha_C(\lambda,\varphi)\right]\lim_{\varphi \to 0}\frac{1}{\tan\varphi}\text{(A18)}$$
This approaches $\pm\infty$ unless the terms associated with the limit of the tangent exactly cancel. This
is achieved if $\alpha_A(\lambda,0°) = \alpha_B(\lambda,0°) = \alpha_C(\lambda,0°)$, which includes the special case where
$\alpha_A(\lambda,0°) = \alpha_B(\lambda,0°) = \alpha_C(\lambda,0°) = 1$. Based on our assumptions of symmetry with the
CSAT3, $\alpha_B(\lambda,\varphi) = \alpha_A(60° - \lambda, -\varphi)$ and $\alpha_C(\lambda,\varphi) = \alpha_A(60° + \lambda, -\varphi)$. Therefore, the $w$
correction for near equatorial winds is unstable unless:
$$\alpha_A(\lambda,0°) = \frac{1+\sqrt{3}\tan\lambda}{2}\alpha_A(60° - \lambda, 0°) + \frac{1-\sqrt{3}\tan\lambda}{2}\alpha_A(60° + \lambda, 0°) \quad \text{(A19)}$$
This is satisfied by $\lambda = 30°$, 90°, 150°, 210°, 270°, and 330°. Eq. A19 shows that if the weighted
average of $\alpha_A(60° - \lambda, -\varphi)$ and $\alpha_A(60° + \lambda, -\varphi)$ cancel $\alpha_A(\lambda,0°)$ then the correction will be
stable. This cannot be achieved if the correction $\alpha_A(\lambda,0°)$ is monotonic between $0° \le \lambda \le 90°$.
Because the $w$ correction is symmetric every 30°, any solution besides $\lambda = 30°$, 90°, 150°, 210°,
270°, and 330° will be mirrored 12 times.



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





Table 1. Summary of the subset of data from Frank et al. (2013) and Frank et al. (2016)
reanalyzed in this study listing the four CSAT3 anemometers (A-D), their location within the
five-position horizontal array, and if mounted horizontally (*). Because processing the Bayesian
model is extremely intensive, only 5% of the available data was reanalyzed.

| | Position | | | | | Number of 5-min periods | |
| Dates | 1 | 2 | 3 | 4 | 5 | Available | Reanalyzed |
|---|---|---|---|---|---|---|---|
| 5-19 July 2011 | A* | B | - | C | D* | 2,520 | 126 |
| 19-26 July 2011 | A | B* | - | C* | D | 1,992 | 100 |
| 9-16 August 2011 | B* | A | - | D | C* | 1,974 | 98 |
| 16-22 August 2011 | B | A* | - | D* | C | 1,620 | 81 |
| 26-30 July 2013 | A* | - | B | - | - | 906 | 46 |
| 23-27 August 2013 | - | - | A | - | B* | 1,050 | 52 |
| 6-24 September 2013 | - | - | B | D* | - | 498 | 25 |






Table 2. Increase in *H* + *LE* (sum of the turbulent components of the energy balance, i.e. sensible
and latent heat flux) at various sites across North America after applying shadow correction to
the CSAT3 time series data.

| Site | Coordinates | Dates | Height (m) | Kaimal correction | Posterior correction mean ± standard deviation [95% credible interval] |
|---|---|---|---|---|---|
| | | | | **Percent change after applying shadow correction** | |
| Yuma, AZ, USA | 33° 5' N 114° 32' W | 6-15 June 2008 | 8.25 | 5.1% | 9.8 ± 2.3% [5.1% 14.8%] |
| Yuma, AZ, USA | 33° 5' N 114° 32' W | 5-14 June 2009 | 2.00 | 4.5% | 9.4 ± 2.8% [3.1% 16.1%] |
| Fraser, CO, USA | 39° 53' 48.23" N 105° 53' 33.87" W | 5-14 April 2015 | 27.50 | 5.6% | 9.9 ± 1.4% [7.4% 12.2%] |
| Fraser, CO, USA | 39° 53' 48.23" N 105° 53' 33.87" W | 5-14 April 2015 | 6.40 | 6.8% | 11.6 ± 1.2% [9.4% 13.9%] |
| Beltsville, MD, USA | 39° 1' 51.23" N 76° 50' 39.40" W | 16-31 July 2014 | 4.00 | 5.5% | 10.4 ± 2.1% [6.3% 14.8%] |
| Glacier Peak, WY, USA | 41° 22' 52" N 106° 15' 47" W | 28 August-8 September 2015 | 3.20 | 5.3% | 11.3 ± 3.1% [4.6% 19.2%] |
| Agua Salud, Panama | 9° 13' 31.65" N 79° 45' 36.41" W | 6-16 November 2015 | 5.00 | 4.7% | 8.1 ± 1.6% [5.3% 10.8%] |




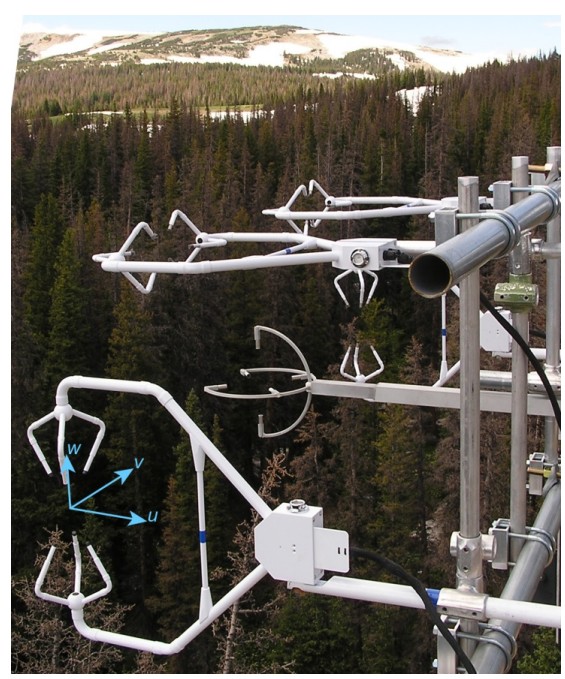


Fig. 1. Photograph of the 2011 experiment with two CSAT3 sonic anemometers mounted
vertically and two horizontally. The cardinal $u$, $v$, and $w$ axes are shown in light blue near one of
the vertical instruments. Figure from Frank et al. (2013).



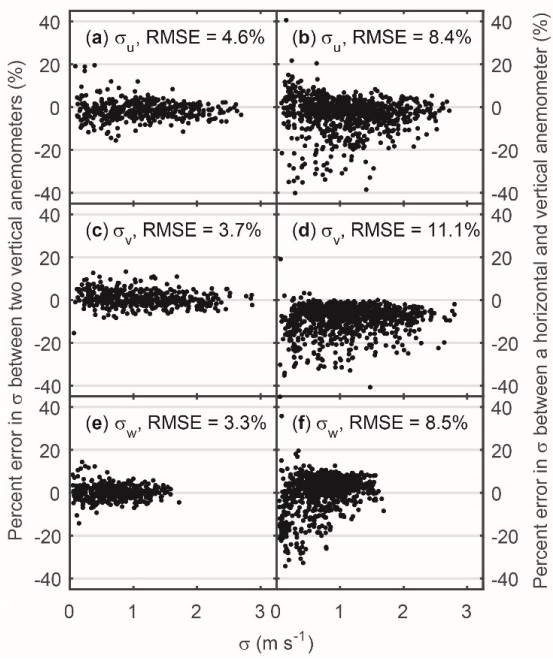


Fig. 2. Uncorrected measurements of the 5-minute standard deviation of wind ($\sigma$) along the
cardinal (**a, b**) $u$, (**c, d**) $v$, and (**e, f**) $w$ axes are not equivalent between vertically and horizontally
mounted CSAT3 sonic anemometers. Data from an ideal 3D anemometer would have similar
percent errors between a horizontal and a vertical anemometer (**b, d, f**) as found between two
anemometers mounted vertically (**a, c, e**). The data are from 2011 and 2013 field experiments at
the GLEES AmeriFlux site (Frank et al., 2016; Frank et al., 2013). The 2011 data in panels **b, d,**
and **f** are randomly paired between the two anemometers in different orientations. Results are
summarized as root mean square error (RMSE).




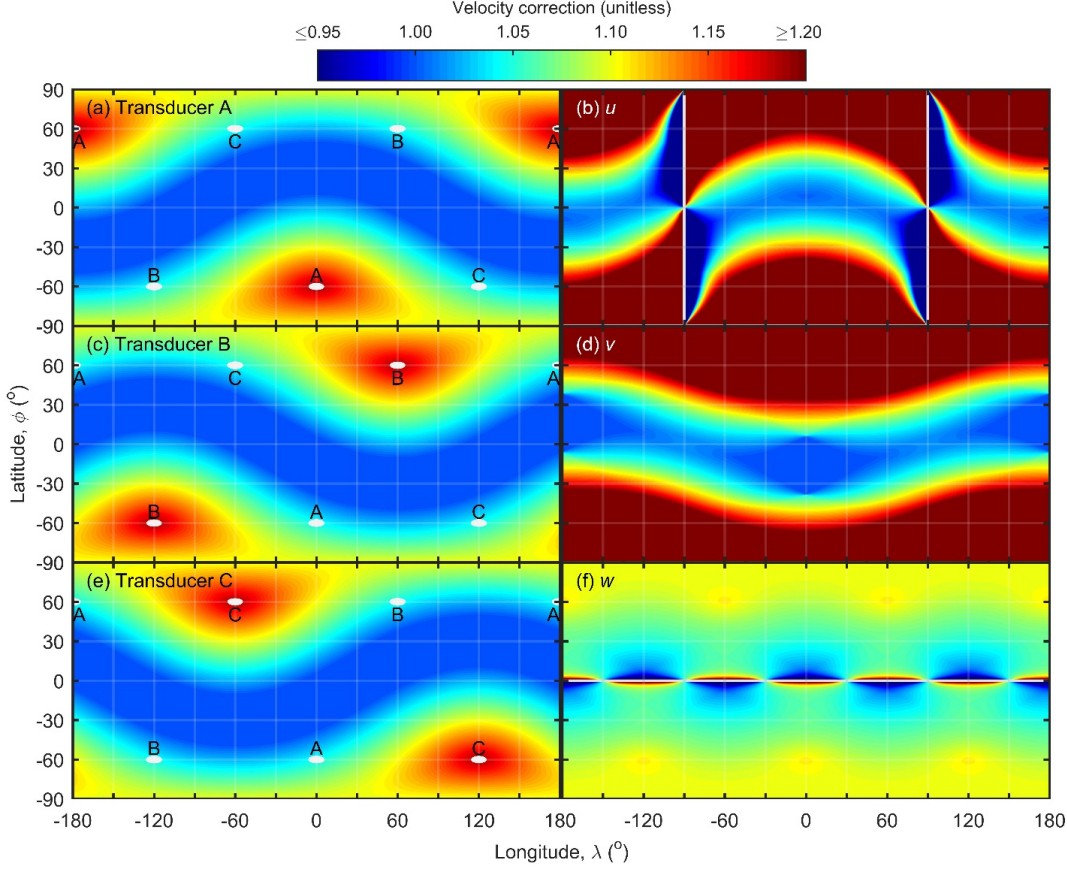


Fig. 3. The Kaimal correction, one of three priors tested in this study, for the (**a**) A, (**c**) B, and (**e**)

C transducer pairs, each represented by a white dot, of a CSAT3 sonic anemometer accounts for

self-shadowing but not cross-shadowing between transducers. The same correction expressed in

sonic anemometer coordinates (**b**) $u$, (**d**) $v$, and (**f**) $w$ shows that for near-equatorial winds,

minimal correction is required for the horizontal wind components while significant correction

and instability exist in the vertical wind component $w$. Longitude and latitude are relative to the $u$

axis (Fig. 1).

825



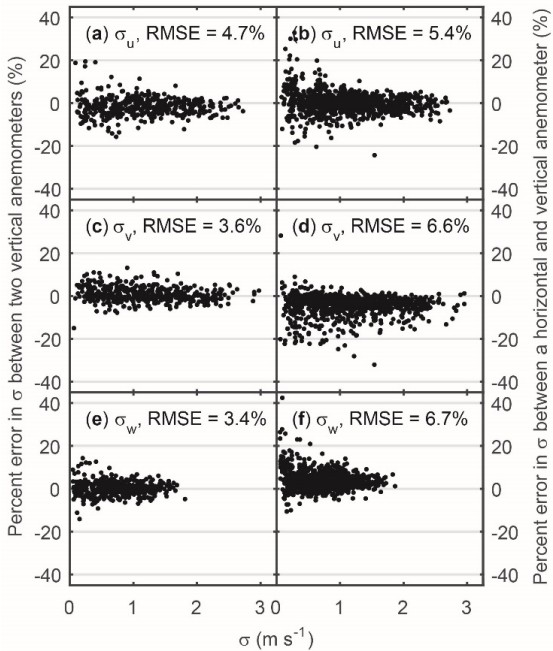

826

Fig. 4. Kaimal corrected measurements (i.e. one of three priors tested) of the 5-minute standard

deviation of wind ($\sigma$) along the cardinal (**a, b**) $u$, (**c, d**) $v$, and (**e, f**) $w$ axes are more equivalent

between vertically and horizontally mounted sonic anemometers. The percent errors between a

horizontal and a vertical anemometer (**b, d, f**) are smaller for all three cardinal dimensions than it

was for the uncorrected data (Fig. 2) being more similar to those found between two

anemometers mounted vertically (**a, c, e**). The data are from 2011 and 2013 field experiments at

the GLEES AmeriFlux site (Frank et al., 2016; Frank et al., 2013). The 2011 data in panels **b, d,**

and **f** are randomly paired between the two anemometers in different orientations. Results are

summarized as root mean square error (RMSE).



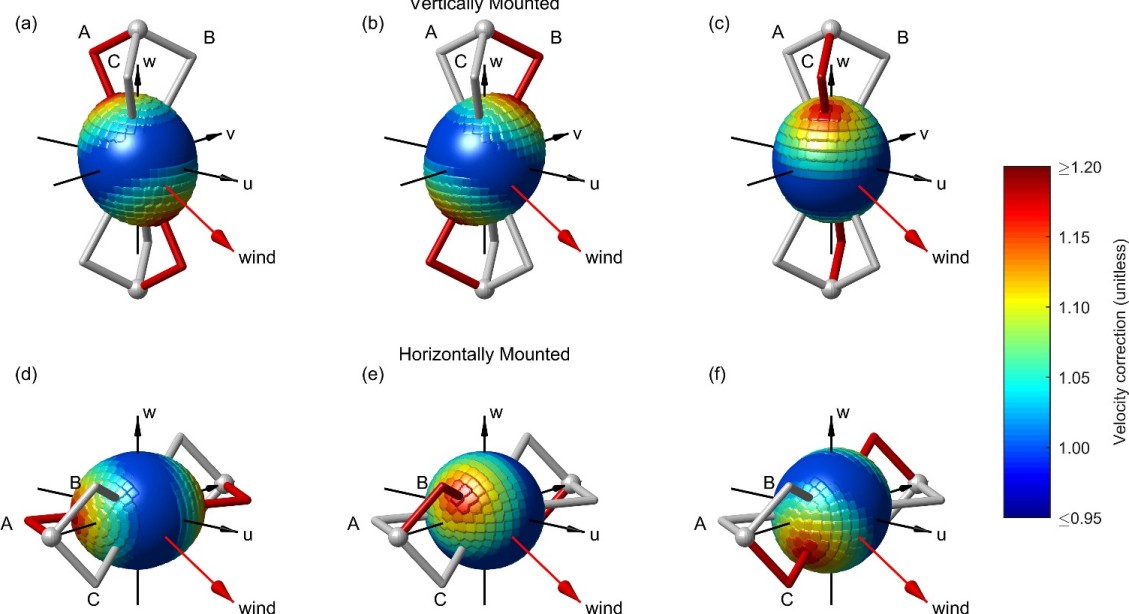


Fig. 5. The Kaimal correction, one of three priors tested in this study, evaluated among 512 cells
for the (**a, d**) A, (**b, e**) B, and (**c, f**) C transducer pairs of the CSAT3 sonic anemometer mounted
either in the (**a-c**) typically vertical or (**d-f**) experimentally horizontal orientations. Though the
correction is identical relative to all transducer pairs, the same instantaneous wind results in
different corrections depending on the transducer pair and the orientation.






Fig. 6. The A transducer pair correction evaluated among 512 cells for the three prior corrections

tested in this study, (**a**) flat, (**c**) Kaimal, and (**e**) double-Kaimal, with their corresponding

unnormalized posterior corrections (**b**), (**d**), and (**f**), respectively. All posteriors have similar

relative topography. They differ in absolute scaling where priors with higher absolute magnitude



result in posteriors with higher absolute magnitude, which is apparent from the different
colorings.



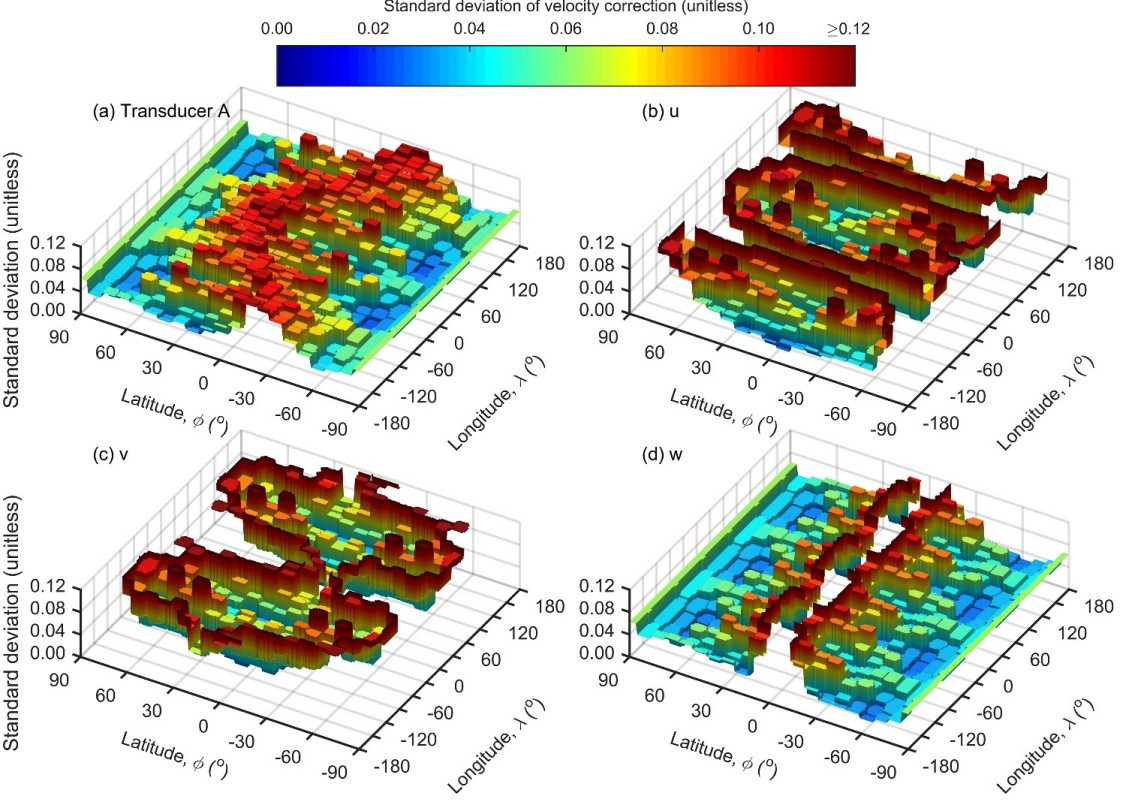


Fig. 7. Standard deviations of the posterior correction for (**a**) the A transducer pair and the wind
velocities (**b**) *u*, (**c**) *v*, and (**d**) *w*. When compared to the standard deviation of the prior which
was defined as 0.1, the transducer correction is more certain in regions with higher topography
(Fig. 6). The results in CSAT3 sonic coordinates reflect both the uncertainty in the transducer
correction plus cancelation and amplification of errors due to the coordinate transformation. The
posterior correction for *u*, *v*, and *w* is most certain for winds along the *u*, *v*, and *w*-axes,
respectively.





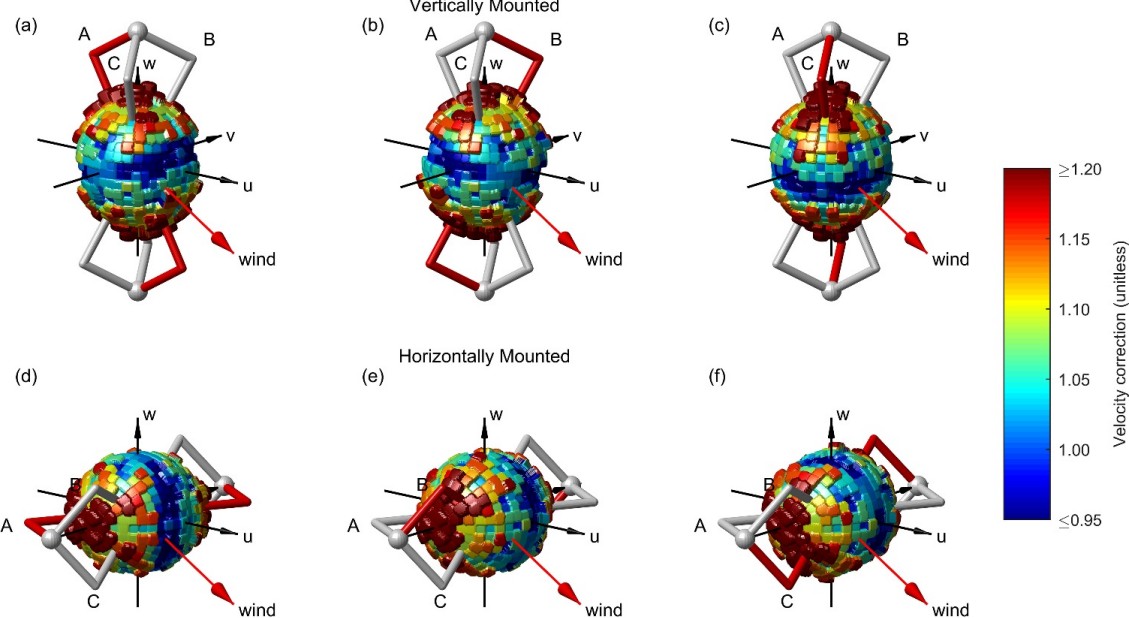


Fig. 8. The posterior correction evaluated for the (**a, d**) A, (**b, e**) B, and (**c, f**) C transducer pairs
of the CSAT3 sonic anemometer mounted either in the (**a-c**) typically vertical or (**d-f**)
experimentally horizontal orientations. The correction is identical relative to all transducer pairs
and is constructed from 512 cells with 138 unique values. The Bayesian model adjusts these
values to simulataneously correct the same instantaneous wind measured from different
transducer pairs and orientations in order to produce similar cardinal *u*, *v*, and *w* wind statistics
(Fig. 10).





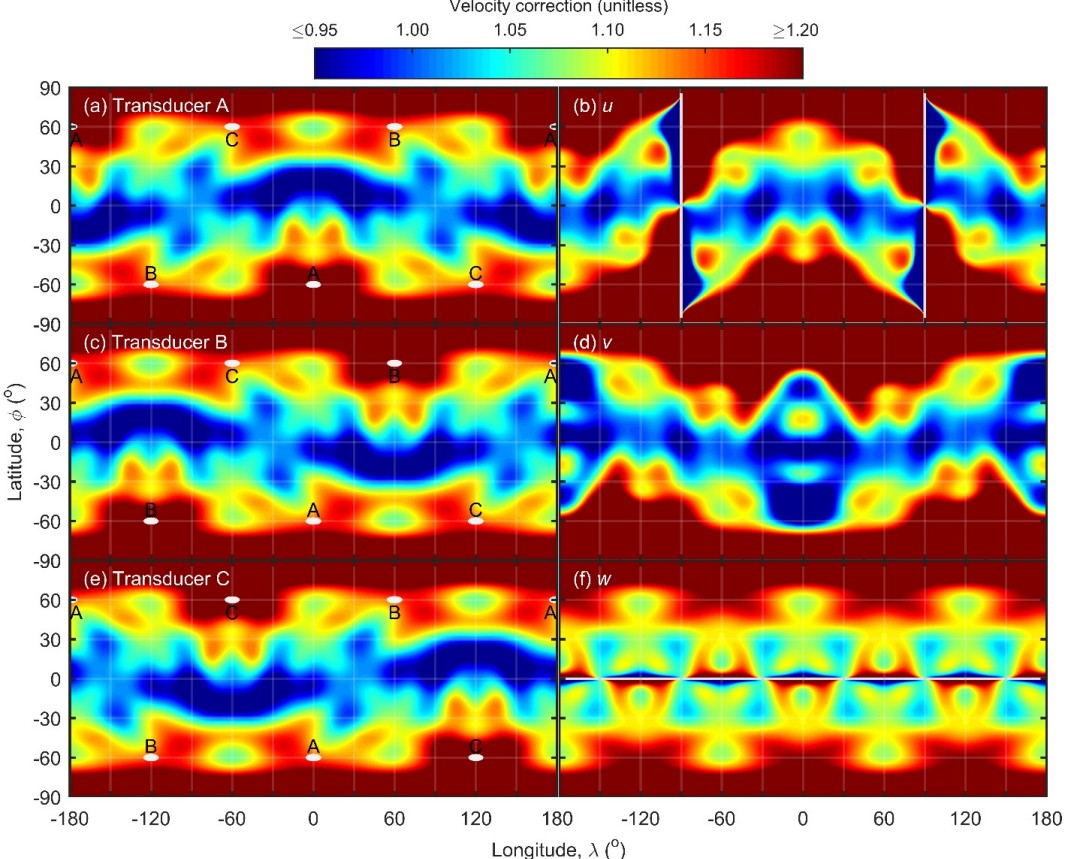

Fig. 9. The posterior correction for the (**a**) A, (**c**) B, and (**e**) C transducer pairs, each represented

by a white dot, of a CSAT3 sonic anemometer accounts for both self-shadowing and cross-

shadowing between transducers. The same correction expressed in sonic anemometer

coordinates (**b**) $u$, (**d**) $v$, and (**f**) $w$ shows that for near-equatorial winds, minimal correction is

required for the horizontal wind components while even more correction exists in the vertical

wind component $w$ than was present with the Kaimal correction (Fig. 3f). Longitude and latitude

are relative to the $u$ axis (Fig. 1).



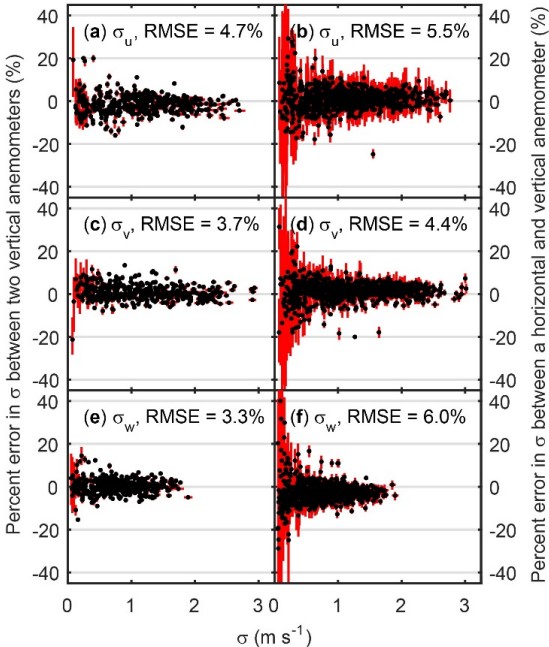


Fig. 10. Posterior corrected measurements of the 5-minute standard deviation of wind ($\sigma$) along
the cardinal (**a, b**) $u$, (**c, d**) $v$, and (**e, f**) $w$ axes are most equivalent between vertically and
horizontally mounted sonic anemometers than with either the uncorrected (Fig. 2) or Kaimal
corrected data (Fig. 4). The percent errors between a horizontal and a vertical anemometer are
small (**b, d, f**), especially for the cardinal $v$-dimension (**d**), and are similar to those found
between two anemometers mounted vertically (**a, c, e**). The data are from 2011 and 2013 field
experiments at the GLEES AmeriFlux site (Frank et al., 2016; Frank et al., 2013). The 2011 data
in panels **b**, **d**, and **f** are randomly paired between the two anemometers in different orientations.
Results are summarized as root mean square error (RMSE). The red lines are 95% credible
intervals.





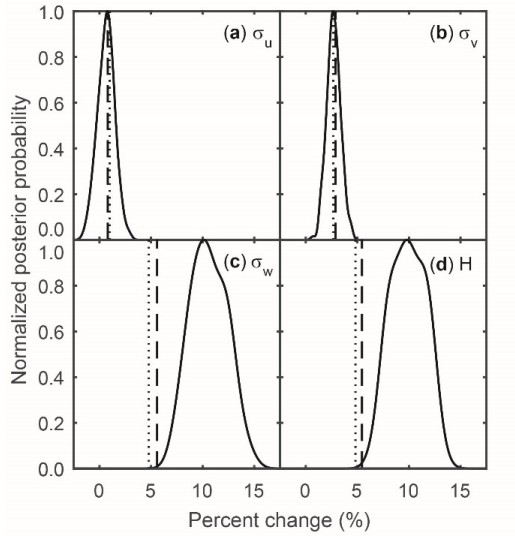


Fig. 11. Though application of the Kaimal (dashed lines) and posterior (solid lines) corrections
result in similar changes to the 5-minute standard deviations of wind ($\sigma$) along the (**a**) *u* and (**b**) *v*
axes, application of the posterior correction results in significantly higher (95% credible interval)
(**c**) winds along the *w* axis and (**d**) sensible heat flux (*H*). The dotted lines are an alternate
formulation of the Kaimal correction proposed by Wyngaard and Zhang (1985) and used in
Horst et al. (2015). Data are for vertically mounted anemometers only.