# Peer review of "A Bayesian model to correct underestimated 3D wind speeds from sonic anemometers"

_Atmospheric Measurement Techniques, 2016_

## Referee Comment (RC1) · Anonymous Referee #2 · 18 Jul 2016

The work presented in "A Bayesian model to correct underestimated 3D wind speeds from sonic anemometers increases turbulent components of the surface energy balance" by J.M. Frank et al. makes a valuable contribution to the improvement and refinement of the eddy-covariance technique for measuring the exchange of mass and energy between the land and atmosphere. The objective of the paper is to show a method which provides a better correction to wind velocities from sonic anemometers for transducer shadowing. The paper is well written and the methods are clearly explained. The three hypotheses laid out by the authors are well substantiated, and throughout the paper the focus on refuting or supporting these objectives is maintained.

The authors showed that by using the 3D correction obtained from the Bayesian approach to high frequency wind velocity data, the measurements from vertically and horizontally mounted anemometers would be more similar. The correction increased vertical wind velocity and sensible heat flux by 10% with a 2% uncertainty. A re-analysis of data from several North American flux sites where the posterior correction was applied to the eddy-covariance data resulted in the turbulent components of the energy balance increasing between 8 to 12%.

Closure of the energy balance in eddy covariance studies can often be problematic, and casts doubt on the mass and energy fluxes derived from the data. Eddy covariance data is also plagued with missing data, which is usually missing not at random, and therefore exacerbates the energy balance closure problem. By reducing sources of systematic bias, it allows investigators to better understand remaining discrepancies in the energy balance. By producing mass and energy fluxes which are closer to the true values, the application of this correction to eddy covariance data would provide better ground-truthing data for land-atmosphere exchange models.

The only draw-back of this analysis carried out on this paper is the small percentage of available data used to derive the posterior correction. As discussed by the authors, the logical next step would be to translate this method to a framework which could make use of parallel computing to speed up the calculations, thereby allowing more data to be used, a larger number of unique corrections, and for the MCMC chains to continue for more steps. If this correction is shown to be stable and consistent when derived from a larger sample of data, and is shown to apply to other anemometers, what would be the proposed uptake of this research? Do the authors expect that the correction will be applied by the firmware of the different anemometers, and therefore current anemometers would then apply the correction to the wind velocity data after a firmware update? Or would it be possible to apply the correction in the post-processing of the raw eddy covariance data, and therefore allow historical data to be re-processed?

---

## Referee Comment (RC2) · Anonymous Referee #3 · 26 Sep 2016

The present manuscript proposes a novel method to correct eddy covariance fluxes from sonic anemometers. It works by jointly estimating "true" standard deviation of components of wind field, a parameter related to the precision of the standard deviation of the un-corrected observations, and a matrix of correction parameters (which contains correction terms for different wind directions).

It is clear that a lot of work went into a paper and the results present a clear improvement over a previously used Kaimal correction. The method is innovative, however it is extremely slow to implement. (This may be solved by potential future numerical

improvements or by an increase in computing power).

Before publication, several points need to be ironed out.

First, no cross-validation of the correction field has been performed. Such a cross-validation is recommended before the method can be generalized to other datasets.

Second, the MCMC chains are very short, even though they use more than a hundred of parameters. Short chains might be prone to misconvergence. In my practice, I needed hundreds of thousands of samples to achieve robust results for around just 10 parameters. Even though the results look similar for different priors, this shortcoming needs to be at least mentioned.

Some of the mathematical notation is confusing; for example the difference between the upper and lower case subscripts need to be better explained. In addition, more attention can be given to explaining the dimensionality of variables (e.g., scalar, matrix, vector).

I encourage the authors to take a final look at the paper to correct some typos. e.g., l. 103 statistics is singular not plural l. 107 data are plural not singular

---

## Author Comment (AC1) · 16 Nov 2016

(1) Comments from Referee

Anonymous Referee 3

The present manuscript proposes a novel method to correct eddy covariance fluxes from sonic anemometers. It works by jointly estimating "true" standard deviation of components of wind field, a parameter related to the precision of the standard deviation of the un-corrected observations, and a matrix of correction parameters (which contains correction terms for different wind directions).

It is clear that a lot of work went into a paper and the results present a clear improvement over a previously used Kaimal correction. The method is innovative, however it is extremely slow to implement. (This may be solved by potential future numerical improvements or by an increase in computing power).

Before publication, several points need to be ironed out.

First, no cross-validation of the correction field has been performed. Such a cross-validation is recommended before the method can be generalized to other datasets.

[see Author response 1]

Second, the MCMC chains are very short, even though they use more than a hundred of parameters. Short chains might be prone to misconvergence. In my practice, I needed hundreds of thousands of samples to achieve robust results for around just 10 parameters. Even though the results look similar for different priors, this shortcoming needs to be at least mentioned.

[see Author response 2]

Some of the mathematical notation is confusing; for example the difference between the upper and lower case subscripts need to be better explained. In addition, more attention can be given to explaining the dimensionality of variables (e.g., scalar, matrix, vector).

[see Author response 3]

I encourage the authors to take a final look at the paper to correct some typos. e.g., l. 103 statistics is singular not plural l. 107 data are plural not singular

[see Author response 4]

(2) Author's response

[Author Response 1]

Though it would be possible to conduct a statistical cross-validation of our model by using the other 95% of the available data, we decided against doing this. First, the 5% of the available data we used was equally distributed from the entire dataset, meaning it is effectively a random subsample and should be representative of any other subset of data selected for validation. We acknowledge that this is only an assumption. Second, as it is, the Bayesian analysis took $\approx 2$ months to complete, thus we decided not to run extra analyses for the purpose of cross-validation. Similar to our response to Anonymous Referee 2, we would have loved to analyze more of the available data, be it for a more extensive estimate of the posterior correction or for cross-validation purposes. But, we have to weigh the potential knowledge gained by analyzing another 5% or 10% of the available data against adding 2-4 months of computer run-time to complete this study.

Taking the reviewer's comments into consideration, we decided to use a different approach to validate the posterior correction by conducting a simple field experiment. The benefit of this is that we could both validate and test the reproducibility of our results. It is a powerful statement if the posterior correction can independently explain similar observations from a different field site, over completely different vegetation, and using different equipment. We also extended our methodology by testing a new manipulation (i.e., askew) which is unique from anything the posterior correction was optimized for. With regards to validation and reproducibility, we contend that this field experiment was a success. The results from the askew test were less definitive, illustrating that there is currently much uncertainty in the posterior correction that needs to be improved by testing more manipulations.

These changes are described in the methods (lines 265-286), results (lines 380-403), discussion (lines 580-613), and table 3.

[Author response 2]

We have added text (lines 204-213) that better describes the tests we used to ensure convergence. We have also mentioned the shortcoming of chains that are too short and could misconvergence.

[Author response 3]

There was also some confusion on our mathematics when our manuscript was initially approved for discussion. We have taken all comments from those reviews plus those from Anonymous Referee 3 into consideration to revise much of section 2.1 and the appendix. We have added a more thorough description of the dimensionality of variables and how they are denoted in lines 130-138.

[Author response 4]

These errors have been corrected along with appropriate grammatical changes to the text.

(3) Author's changes in manuscript

(The following are excerpted from the revised manuscript, which is included as a supplemental material)

[lines 130-138]

In our mathematical notation, we use uppercase and lowercase subscripts to distinguish variables as scalars, vectors, or matrices. Uppercase subscripts are part of the variable name and denote the dimensionality of the variable as well as describe the coordinate system. For example, $M_{S \times T}$ is a two dimensional matrix with dimensions S and T, which correspond to sonic and transducer coordinates; since there are three dimensions for both coordinate systems this is a 3x3 matrix. One uppercase subscript by itself denotes a vector in that coordinate system. Lowercase subscripts denote indexing for variables that are defined for multiple times or replicate anemometers; these

are essentially multidimensional arrays. When the same letter appears as both an uppercase and lowercase subscript, this refers to the $c^{th}$ element of dimension C.

[lines 204-213]

We conducted several preliminary Bayesian analyses, and used trace plots and tests for autocorrelation to determine that 10,000 steps was sufficient for convergence for most of the 138 state variables defining $\alpha_T$. Most of these parameters required little or no thinning to reduce autocorrelation between steps and could have remained as MCMC chains with 1,000-10,000 steps. Yet, since the goal was to create a complete 3D correction, we decided to thin all state variables equally. Even though diagnostic tests showed that all parameters, including those with high autocorrelation, appeared to converge within 10,000 steps, it is possible that these chains are still too short for proper convergence. One safeguard against this is confirming that the results from the three chains all result in similar posterior distributions (see section 3.3).

[lines 265-286]

2.4 Validation experiment

We conducted a validation experiment of the posterior 3D correction at the Colorado State University, Agricultural Research Development and Education Center (ARDEC), Fort Collins, CO, USA (40° 39' 7.9" N 104° 59' 45.7" W) from October 7-14, 2016 . Three CSAT3 sonic anemometers were mounted on an east-west boom 2 m above a pasture of short grass and ≈36 m south of a mature corn field. Typical winds at this site are from the north, so in this experiment we refer to cardinal u, v, and w where the measurements have been rotated to north-south (u), west-east (v), and down-up (w). One anemometer (S/N 0869) was vertically mounted in the center of the boom and aimed north, a second (S/N 1560) was 0.62 m to the east and horizontally mounted (i.e., 90° rotation around its u-axis) and aimed north, and a final instrument (S/N 2385) was 0.58 m to the west and mounted askew (Fig. S1). The askew mounting is unique to this validation experiment and can be defined with the unit vectors u (pointing south),

v (pointing east), and w (pointing up) as $u_{askew}$ = 2/3u - 1/3v - 2/3w, $v_{askew}$ = 2/3u + 2/3v + 1/3w, and $w_{askew}$ = 1/3u − 2/3v + 2/3w. All wind velocity measurements were converted from sonic to cardinal coordinates, and all tilt angles were measured with a digital level to 0.1° precision such that any mounting imperfections were taken into account. Data were measured at 20 Hz on a CR3000 micrologger (Campbell Scientific, Inc.). In post processing, both the Kaimal correction and the posterior 3D correction were applied to the 20 Hz data. Data were summarized every 5 minutes as the standard deviation of wind velocity along the cardinal directions, $\sigma_u$, $\sigma_v$, and $\sigma_w$. Differences between anemometers are presented as root-mean-square of the relative error (RMSE) between measurements from the manipulated anemometers and the vertically mounted one.

[lines 380-403]

3.6 Validation of the posterior correction

The validation experiment was conducted during excellent fall weather with no precipitation, where winds averaged 2.0 ± 1.2 m s$^{-1}$, maximum sustained gusts were 7.8 m s$^{-1}$, 38% of the winds were from the northeast (45°) to north-northwest (337.5°), 25% of the winds were from the southeast (135°) to south (180°), and during the other times there were some occasional westerly winds. Results are summarized in Table 3. The RMSE differences between a horizontally mounted anemometer and a vertically mounted anemometer were large (12.6-16.5%) for uncorrected measurements. Applying the Kaimal correction to these anemometers reduced the RMSE differences in $\sigma_u$ and $\sigma_v$ (8.5 and 11.4%) but increased the difference in $\sigma_w$, (17.5%). Compared to the uncorrected data, the average posterior correction decreased the RMSE differences in all directions, though only the reduction in $\sigma_v$ (8.0-12.2%) was statistically lower (i.e., 95% credible interval). Compared to the Kaimal correction, the average posterior correction was larger for $\sigma_u$ but lower for $\sigma_v$ and $\sigma_w$, with the reduction in $\sigma_w$ (11.8-15.9%) being statistically lower than with the Kaimal corrected data. The RMSE differences between an askew mounted anemometer and a vertically mounted anemometer were

small/moderate for $\sigma_u$ and $\sigma_v$ (6.7% and 11.3%) and large for $\sigma_w$ (14.7%) for uncorrected measurements. Applying the Kaimal correction to these anemometers reduced the RMSE differences in all directions (4.4-13.5%). The standard deviations for the RMSE differences using the posterior correction was higher for the askew manipulation (1.5-2.4%) than they were for the horizontal manipulation (1.1-1.3%). Compared to the uncorrected data, the average posterior correction increased the RMSE difference for $\sigma_u$ (8.6%) but decreased the differences for $\sigma_v$ and $\sigma_w$ (10.3% and 13.9%), though none of these changes were statistically significant. Compared to the Kaimal correction, the average posterior correction increased the RMSE differences for all directions, with the differences in $\sigma_u$ (6.2-11.6%) and $\sigma_v$ (7.2-13.5%) being statistically larger.

[lines 580-613]

Sonic anemometer corrections should be verified and validated. There is an opportunity to statistically cross-validate the posterior 3D correction with subsets of the other 95% of available data; we decided against this because the 5% used was already partitioned equally throughout the full dataset, plus, analyzing multiple rounds of training and validation datasets would take additional months of computation. Instead of a statistical cross-validation analysis, we conducted a validation field experiment to determine if (1) our results are reproducible and (2) if they can explain other manipulations. From this, we first conclude that our results are reproducible. In both our main experiments at GLEES and the validation experiment at ARDEC, there was improved agreement between vertically and horizontally mounted anemometers when using the posterior correction versus the Kaimal correction or no-correction (Table 3). The largest differences between anemometers was for $\sigma_v$ (11.1% and 16.5%, Fig. 2d, Table 3) which were reduced with the Kaimal correction (6.6% and 11.4%, Fig. 4d, Table 3) and then further improved with the posterior correction (4.4% and 9.8%, Fig. 10d, Table 3). In both analyses, the differences in $\sigma_u$ were reduced with either correction, but the best performance was the Kaimal prior (Figs. 4b versus 10b, Table 3). Finally, in both cases the differences in $\sigma_w$, were smallest using the posterior correction (Figs. 4f versus

10f, Table 3). Moreover, we justify our validation because it involved an independent dataset that was collected at a different field site, over radically different terrain and vegetation, and using anemometers with different serial numbers. We are less confident that our posterior correction can explain all manipulations. The differences in $\sigma_u$ and $\sigma_v$ between vertically and askew mounted anemometers were significantly better with the Kaimal correction (Table 3). It is important to note, however, that these differences were the smallest of all the comparisons (Uncorrected column in Table 3); i.e., it may be inconsequential that the Kaimal correction outperforms the posterior correction for measurements that were fairly good to begin with. Meanwhile, the difference in $\sigma_w$, was large, though it is unclear if the posterior correction makes this significantly better or worse (Table 3). This lack of clarity means the askew manipulation cannot be used to validate or falsify the posterior correction. This is not surprising, because the posterior correction was estimated without data from or knowledge of such a unique manipulation, and as it is, much of the posterior correction contains a large uncertainty (Fig. 7a). Though the posterior correction is too uncertain to explain the askew manipulation, this does not mean our estimates of H + LE at various field sites are flawed because these estimates account for the fact that much of the posterior is uncertain. We expect that expanding our Bayesian analysis to include data from more manipulations, e.g. the askew example, would further constrain the regions of uncertainty found in the current posterior correction.

[Table 3]

(see supplemental pdf for Table 3)

Please also note the supplement to this comment:
http://www.atmos-meas-tech-discuss.net/amt-2016-145/amt-2016-145-AC1-supplement.pdf

---

## Author Comment (AC2) · 17 Nov 2016

(1) Comments from Referee

Anonymous Referee 2

The work presented in "A Bayesian model to correct underestimated 3D wind speeds from sonic anemometers increases turbulent components of the surface energy balance" by J.M. Frank et al. makes a valuable contribution to the improvement and refinement of the eddy-covariance technique for measuring the exchange of mass and energy between the land and atmosphere. The objective of the paper is to show a

method which provides a better correction to wind velocities from sonic anemometers for transducer shadowing. The paper is well written and the methods are clearly explained. The three hypotheses laid out by the authors are well substantiated, and throughout the paper the focus on refuting or supporting these objectives is maintained.

The authors showed that by using the 3D correction obtained from the Bayesian approach to high frequency wind velocity data, the measurements from vertically and horizontally mounted anemometers would be more similar. The correction increased vertical wind velocity and sensible heat flux by 10% with a 2% uncertainty. A re-analysis of data from several North American flux sites where the posterior correction was applied to the eddy-covariance data resulted in the turbulent components of the energy balance increasing between 8 to 12%.

Closure of the energy balance in eddy covariance studies can often be problematic, and casts doubt on the mass and energy fluxes derived from the data. Eddy covariance data is also plagued with missing data, which is usually missing not at random, and therefore exacerbates the energy balance closure problem. By reducing sources of systematic bias, it allows investigators to better understand remaining discrepancies in the energy balance. By producing mass and energy fluxes which are closer to the true values, the application of this correction to eddy covariance data would provide better ground-truthing data for land-atmosphere exchange models.

The only draw-back of this analysis carried out on this paper is the small percentage of available data used to derive the posterior correction. As discussed by the authors, the logical next step would be to translate this method to a framework which could make use of parallel computing to speed up the calculations, thereby allowing more data to be used, a larger number of unique corrections, and for the MCMC chains to continue for more steps. If this correction is shown to be stable and consistent when derived from a larger sample of data, and is shown to apply to other anemometers, what would be the proposed uptake of this research? Do the authors expect that the correction will be applied by the firmware of the different anemometers, and therefore current

anemometers would then apply the correction to the wind velocity data after a firmware update? Or would it be possible to apply the correction in the post-processing of the raw eddy covariance data, and therefore allow historical data to be re-processed?

(2) Author's response

For now, we anticipate this specific posterior correction will be used for sensitivity tests and error analysis during post-processing of raw eddy covariance data. We deliberately made it available as a supplemental file to encourage users to investigate the potential for how much transducer shadowing could influence their understanding of physical processes at their field sites. We also encourage development of the Bayesian model to obtain better resolution in the correction and to extend it to include different manipulations and anemometers, etc. We are disappointed that we could only analyze 5% of our available data, and we look forward to developments that adapt our model to high performance computing and parallel processing. There may become a time when a 3D shadowing correction is produced that would be appropriate for inclusion in instrument firmware. If that time should come, there would need to be a discussion among manufacturers and the community on how to adapt Bayesian uncertainties to the sensor output.

A problem hindering all research into sonic anemometer errors is there is no consensus on what is an appropriate standard to validate against. This was discussed recently at the AmeriFlux meeting. We are encouraged that using an omnidirectional standard in conjunction with experimental manipulations within a Bayesian framework might provide a framework to answer this tough question. Yet, this study is merely the first step. In response to Anonymous Referee 3, we conducted a brief field validation experiment where we tested a completely new manipulation (i.e. askew). While we were disappointed that the results for this manipulation were fairly ambiguous, we interpreted this to mean that there is a wealth of information yet to be incorporated into this type of analysis.

(3) Author's changes in the manuscript

There were no changes to the manuscript in response to Referee 2
* * *